# On Batch Teaching with Sample Complexity Bounded by VCD

**Farnam Mansouri**
University of Toronto
frnm.mansouri@gmail.com

**Hans U. Simon**
Max Planck Institute for Informatics
hsimon@mpi-inf.mpg.de

**Adish Singla**
Max Planck Institute for Software Systems
adishs@mpi-sws.org

**Sandra Zilles**
University of Regina
zilles@cs.uregina.ca

## Abstract

In machine teaching, a concept is represented by (and inferred from) a small number of labeled examples. Various teaching models in the literature cast the interaction between teacher and learner in a way to obtain a small complexity (in terms of the number of examples required for teaching a concept) while obeying certain constraints that are meant to prevent unfair collusion between teacher and learner. In recent years, one major research goal has been to show interesting relationships between teaching complexity and the VC-dimension (VCD). So far, the only interesting relationship known from batch teaching settings is an upper bound quadratic in the VCD, on a parameter called recursive teaching dimension. The only known upper bound on teaching complexity that is linear in VCD was obtained in a model of teaching with sequences rather than batches. This paper is the first to provide an upper bound of VCD on a batch teaching complexity parameter. This parameter, called $\mathrm{STD_{min}}$, is introduced here as a model of teaching that intuitively incorporates a notion of "importance" of an example for a concept. In designing the $\mathrm{STD_{min}}$ teaching model, we argue that the standard notion of collusion-freeness from the literature may be inadequate for certain applications; we hence propose three desirable properties of teaching complexity and demonstrate that they are satisfied by $\mathrm{STD_{min}}$.

## 1  Introduction

The notion of machine teaching refers to the selection of helpful training examples that aid the learner in identifying a target concept [Zhu et al., 2018]. Such processes can make machine learning feasible or economical in situations when it is difficult or expensive to acquire large amounts of training data. Human-in-the-loop settings [Wang et al., 2020] or inverse reinforcement learning [Kamalaruban et al., 2019] are potential application areas of machine teaching. Formal studies of teaching consider various teaching models, each of which has a corresponding complexity parameter referring to the worst-case number of examples required to teach any concept in a given concept class [Goldman and Kearns, 1995, Shinohara and Miyano, 1991, Zilles et al., 2011, Gao et al., 2017b, Kirkpatrick et al., 2019, Mansouri et al., 2019]. Each model has to adhere to rules that prevent teacher and learner from using unwanted "coding tricks", such as, for instance, agreeing on an indexing of concepts and an indexing of data points, and then allowing the teacher to simply teach the concept indexed $i$ using the data point with index $i$. While no general notion of coding trick is agreed upon in the literature, Goldman and Mathias [1996] provided a notion of collusion-freeness (which we call GM-collusion-freeness for short) that is often used as an underlying constraint in teaching models.

36th Conference on Neural Information Processing Systems (NeurIPS 2022).

One major line of research addresses the question of how teaching complexity parameters relate to one another and to other parameters of interest in computational learning theory, such as, for instance, the VC-dimension (VCD, Vapnik and Chervonenkis, 1971), the self-directed learning complexity [Goldman and Sloan, 1994], the optimal mistake bound from online learning [Littlestone, 1988], or the size of sample compression schemes [Littlestone and Warmuth, 1986]. Of particular interest is the question how teaching complexity relates to the complexity of learning from randomly chosen examples, and thus to the VCD [Blumer et al., 1989]. Answers to this question provide insights into structural properties that make a concept class easy or hard to teach, and have implications on the design and analysis of teaching and learning algorithms. For example, it was established that the Recursive Teaching Dimension (RTD, Zilles et al., 2011) is upper-bounded by a function quadratic in VCD [Hu et al., 2017]. While RTD may exceed VCD, it remains open whether it is in $O(\text{VCD})$. Connections between RTD and the size of an optimal sample compression scheme were also made [Doliwa et al., 2014, Darnstädt et al., 2016]. More recently, Kirkpatrick et al. [2019] introduced the No-Clash Teaching Dimension (NCTD), which is the optimal complexity of teachers satisfying GM-collusion-freeness. Since the RTD model also satisfies GM-collusion-freeness, NCTD is upper-bounded by RTD and thus trivially inherits Hu et al.'s upper bound quadratic in VCD. It remains unresolved whether or not $\text{NCTD} \leq \text{VCD}$ holds in general; no counter-example has been found yet.

The models of RTD and NCTD represent teaching with batches (sets of examples). Naturally, a smaller complexity can be obtained when teachers can encode concepts in sequences of examples. So far, the only known teaching complexity upper-bounded by VCD stems from a sequential model [Mansouri et al., 2019]. The main problem we address in this paper is to design a reasonable batch model of teaching that has a complexity upper-bounded by VCD.

While GM-collusion-freeness is often considered a very natural condition on teaching, it has recently been argued that there are application settings in which this condition is *not* natural [Ferri et al., 2022]. In addition, since the question whether $\text{NCTD} \leq \text{VCD}$ is not yet resolved, and NCTD is the optimal complexity of GM-collusion-free teaching, our approach is to consider batch models that do not satisfy GM-collusion-freeness, but instead satisfy minimum postulates of teaching without "coding tricks". Our first contribution is to formally define three such postulates and to analyze some teaching models from the literature with respect to these postulates. We then define a new batch teaching model that is not GM-collusion-free but provably satisfies our three postulates of teaching without coding tricks. Our main results are that the corresponding complexity parameter is upper-bounded by both RTD and VCD. This makes our paper the first to provide an upper bound of VCD or $O(\text{VCD})$ on a complexity parameter for batch teaching.

All proofs or parts of proofs omitted from the paper are provided in the appendix.

## 2   Preliminaries

Throughout this paper, the symbol $X$ is used to denote any finite domain over which concepts, i.e., subsets of $X$, are defined. A *concept class* over $X$ is then a finite set of concepts over $X$. In the literature, any kind of finite or infinite set system can serve as a concept class; examples of finite concept classes are given in the form of tables throughout this paper. Often the concept classes studied are highly structured classes of Boolean functions, such as, for instance the class of all monomials over a fixed set of variables [Zilles et al., 2011]. Infinite classes of geometric concepts, such as hyperplanes in the Euclidean space, are often studied as concept classes as well; see, e.g., [Gao et al., 2017a,b].

If $C$ is any concept over $X$ and $x \in X$, then $C(x)$ is the label of $C$ for $x$, i.e., $C(x) = 1$ if $x \in C$, while $C(x) = 0$ if $x \notin C$. A *labeled example* for $C$ is a pair $(x, C(x))$ for $x \in X$. As this paper focuses on batch teaching rather than sequential teaching, teachers and learners are designed to operate with sets of labeled examples. Any set of labeled examples for $C$ is called a *sample set* for $C$. If $\mathcal{C}$ is a concept class, $C \in \mathcal{C}$, and $S$ is a sample set for $C$, we say that $S$ is a *teaching set* for $C$ (wrt $\mathcal{C}$) if it is not a sample set for any $C' \neq C$, $C' \in \mathcal{C}$. A *teacher* for $\mathcal{C}$ is a mapping $T$ that assigns to every concept $C \in \mathcal{C}$ a sample set for $C$. Its *order* $\text{ord}(T)$ is the size of the largest set in its range, i.e., $\text{ord}(T) = \max_{C \in \mathcal{C}} |T(C)|$. A *learner* is a (not necessarily total) mapping that assigns a concept to each sample set in its domain. A learner $L$ and a teacher $T$ are said to *match* on a concept class $\mathcal{C}$, if $L(T(C)) = C$ for every concept $C \in \mathcal{C}$. If $\mathcal{C}$ is a concept class over $X$ and $X' \subseteq X$ is any subset of the domain, then we use $\mathcal{C}{\downarrow}_{X'}$ to denote the *restriction* of $\mathcal{C}$ to $X'$, i.e.,

$\mathcal{C}{\downarrow}_{X'} = \{C{\downarrow}_{X'} \mid C \in \mathcal{C}\}$, where $C{\downarrow}_{X'}$ is the projection of $C$ onto $X'$. We say that $X'$ *preserves* $\mathcal{C}$ if $C_1 \neq C_2$ implies $C_1{\downarrow}_{X'} \neq C_2{\downarrow}_{X'}$ whenever $C_1, C_2 \in \mathcal{C}$. A concept class over $X$ is *succinct* if it is not preserved by any proper subset of $X$.

We now define various notions of batch teaching that have been studied in the literature, beginning with the classical notion of teaching dimension.

**Definition 1 (Shinohara and Miyano [1991], Goldman and Kearns [1995])** *Let $\mathcal{C}$ be a concept class over a domain $X$ and $C \in \mathcal{C}$ be a concept. The teaching dimension of $C$ in $\mathcal{C}$, denoted by $\mathrm{TD}(C, \mathcal{C})$, is the size of the smallest teaching set for $C$ wrt $\mathcal{C}$. The Teaching Dimension of $\mathcal{C}$ is defined as $\mathrm{TD}(\mathcal{C}) := \max\{\mathrm{TD}(C, \mathcal{C}) \mid C \in \mathcal{C}\}$. Moreover, $\mathrm{TD}_{\min}(\mathcal{C}) := \min\{\mathrm{TD}(C, \mathcal{C}) \mid C \in \mathcal{C}\}$. A $\mathrm{TD}$-teacher for $\mathcal{C}$ is a teacher that maps any $C \in \mathcal{C}$ to a smallest teaching set for $C$.*

A TD-teacher matches every learner that outputs concepts consistent with the given sample set. The more recent literature proposes models in which "smarter" learners can be used. In recursive teaching, one exploits a canonical partial order over the concepts in the class: first, the concepts with smallest teaching dimension wrt $\mathcal{C}$ are encoded with their minimum teaching sets, then one removes these concepts from $\mathcal{C}$ and proceeds recursively with the remaining concepts, always encoding and removing the concepts that are "easiest to teach".

**Definition 2 (Zilles et al. [2011])** *Let $\mathcal{C}$ be a concept class over a domain $X$. Let $\mathcal{C}_0 = \mathcal{C}$ and recursively define $\mathcal{C}_i^{\min} = \{C \in \mathcal{C}_i \mid \mathrm{TD}(C, \mathcal{C}_i) = \mathrm{TD}_{\min}(\mathcal{C}_i)\}$ as well as $\mathcal{C}_{i+1} = \mathcal{C}_i \setminus \mathcal{C}_i^{\min}$. An $\mathrm{RTD}$-teacher for $\mathcal{C}$ is a teacher $T$ with the following property: for $C \in \mathcal{C}_i^{\min}$, the set $T(C)$ is a teaching set for $C$ wrt $\mathcal{C}_i$. Then the Recursive Teaching Dimension of $\mathcal{C}$ is defined as $\mathrm{RTD}(\mathcal{C}) := \max\{\mathrm{TD}_{\min}(\mathcal{C}_i) \mid i \geq 0\}$, which equals the smallest possible order of any $\mathrm{RTD}$-teacher for $\mathcal{C}$. A recursive teaching plan for $\mathcal{C}$ is any sequence*

$$\left((C_{1,1}, T(C_{1,1})), \ldots, (C_{1,n_1}, T(C_{1,n_1})), \ldots, (C_{k,1}, T(C_{k,1})) \ldots, (C_{k,n_k}, T(C_{k,n_k}))\right)$$

*where $T$ is an $\mathrm{RTD}$-teacher, $\mathcal{C}_i^{\min} = \{C_{i,1}, \ldots, C_{i,n_i}\}$ for all $i$, and $\mathcal{C} = \mathcal{C}_1^{\min} \cup \ldots \cup \mathcal{C}_k^{\min}$.*

For finite concept classes, the notion of RTD coincides with the Preference-Based Teaching Dimension (PBTD) [Gao et al., 2017b], so our results will similarly apply to PBTD.

No-clash teaching is strictly more powerful than recursive teaching; here the only constraint on the teacher is that no two concepts agree on the union of the two sample sets encoding them.

**Definition 3 (Kirkpatrick et al. [2019])** *Let $\mathcal{C}$ be a concept class over a domain $X$ and $T$ be a teacher for $\mathcal{C}$. We call $T$ a non-clashing teacher ($\mathrm{NCTD}$-teacher) on $\mathcal{C}$ iff there are no two distinct $C, C' \in \mathcal{C}$ such that $T(C)$ is a sample set for $C'$ and $T(C')$ is a sample set for $C$. The No-Clash Teaching Dimension of $\mathcal{C}$ is defined as $\mathrm{NCTD}(\mathcal{C}) := \min\{\mathrm{ord}(T) \mid T \text{ is an } \mathrm{NCTD}\text{-teacher on } \mathcal{C}\}$.*

It is not hard to show that $\mathrm{TD} \geq \mathrm{RTD} \geq \mathrm{NCTD}$ [Kirkpatrick et al., 2019]. A batch teaching parameter that is incomparable to both RTD and NCTD is the so-called subset teaching dimension. In subset teaching, one considers all minimum-size teaching sets for each concept, and iteratively reduces them to minimum-size subsets that are not contained in any of the sets assigned to other concepts. The underlying idea is that a learner can recognize a concept even by a subset $S$ of a smallest teaching set if $S$ is not a subset of a set used for teaching another concept.

**Definition 4 (Zilles et al. [2011])** *Let $\mathcal{C}$ be a concept class over a domain $X$, and $C \in \mathcal{C}$. Define $\mathrm{STS}^0(C, \mathcal{C}) := \{\{(x, C(x)) \mid x \in X\}\}$. For $k \in \mathbb{N}$, iteratively define a collection $\mathrm{STS}^{k+1}(C, \mathcal{C})$ of subset teaching sets as the collection that contains all smallest-size sets $S$ that satisfy the following:*

  *1. $S \subseteq S'$ for some $S' \in \mathrm{STS}^k(C, \mathcal{C})$;*

  *2. $S \nsubseteq S'$ for all $S' \in \mathrm{STS}^k(C', \mathcal{C})$ where $C' \in \mathcal{C}$, $C' \neq C$.*

*Let $k^*$ be minimal with $\mathrm{STS}^k(\hat{C}, \mathcal{C}) = \mathrm{STS}^{k^*}(\hat{C}, \mathcal{C})$ for all $k > k^*$, $\hat{C} \in \mathcal{C}$. An $\mathrm{STD}$-teacher for $\mathcal{C}$ is any teacher $T$ for $\mathcal{C}$ with $T(\hat{C}) \in \mathrm{STS}^{k^*}(\hat{C}, \mathcal{C})$ for all $\hat{C} \in \mathcal{C}$. The Subset Teaching Dimension of $\mathcal{C}$, denoted by $\mathrm{STD}(\mathcal{C})$, is defined as $\mathrm{STD}(\mathcal{C}) := \mathrm{ord}(T)$ where $T$ is any $\mathrm{STD}$-teacher for $\mathcal{C}$.*

STD is well-defined since every STD-teacher for a class $\mathcal{C}$ has the same order, with all sets in $\text{STS}^{k^*}(C, \mathcal{C})$ being of the same size for any fixed $C$. Note that $\text{STS}^1(C, \mathcal{C})$ consists of all minimum-size teaching sets for $C$ wrt $\mathcal{C}$. In general, $\text{TD} \geq \text{STD}$, while STD and RTD are incomparable and either one of them can be larger than the other by an arbitrary factor [Zilles et al., 2011].

These teaching notions are illustrated with the help of Table 1, which shows a concept class $\mathcal{C}_3^{pair}$ with 16 concepts over a domain of size 11. We claim that $\text{TD}(\mathcal{C}_3^{pair}) = 4$, $\text{RTD}(\mathcal{C}_3^{pair}) = 3$, $\text{NCTD}(\mathcal{C}_3^{pair}) = 2$, $\text{STD}(\mathcal{C}_3^{pair}) = 1$, and $\text{TD}_{\min}(\mathcal{C}_3^{pair}) = 1$. Note that the smallest teaching set for any even-numbered concept $C_{2i}$ contains only one example, namely for the instance $x_i$; thus $\text{TD}_{\min}(\mathcal{C}_3^{pair}) = 1$. By contrast, the smallest teaching set for any odd-numbered concept $C_{2i-1}$ contains labeled examples for the instances $x_i$, $x_9$, $x_{10}$, and $x_{11}$. The last three instances are needed to distinguish $C_{2i-1}$ from all other odd-numbered concepts, and $x_i$ is needed to distinguish $C_{2i-1}$ from $C_{2i}$. Hence $\text{TD}(\mathcal{C}_3^{pair}) = 4$. To see that $\text{RTD}(\mathcal{C}_3^{pair}) = 3$, observe that a recursive teaching plan will first list all even-numbered concepts (whose TD is 1), and then has only the odd-numbered ones left, which correspond to the powerset on $\{x_9, x_{10}, x_{11}\}$; clearly these will require a teacher to use sets of size 3. A non-clashing teacher of order 2 is indicated by the boldface labels in the table. One can argue (see the proof of Proposition 17) that no non-clashing teacher of smaller order exists, i.e., $\text{NCTD}(\mathcal{C}_3^{pair}) = 2$. Finally, $\text{STD}(\mathcal{C}_3^{pair}) = 1$ was shown by Zilles et al. [2011]; in the first iteration of subset teaching, even-numbered concepts are assigned their unique teaching sets of size 1, while odd-numbered concepts $C_{2i-1}$ are assigned four-element teaching sets containing the example $(x_i, 0)$. Since the latter is not contained in a minimum teaching set of any other concept, in the next iteration, $C_{2i-1}$ is assigned the subset teaching set $\{(x_i, 0)\}$.

| concept in $\mathcal{C}_3^{pair}$ | $x_1$ | $x_2$ | $x_3$ | $x_4$ | $x_5$ | $x_6$ | $x_7$ | $x_8$ | $x_9$ | $x_{10}$ | $x_{11}$ |
|---|---|---|---|---|---|---|---|---|---|---|---|
| $C_1$ | **0** | 0 | 0 | 0 | 0 | 0 | 0 | 0 | **0** | **0** | 0 |
| $C_2$ | **1** | 0 | 0 | 0 | 0 | 0 | 0 | 0 | 0 | 0 | 0 |
| $C_3$ | 0 | **0** | 0 | 0 | 0 | 0 | 0 | 0 | **0** | 0 | **1** |
| $C_4$ | 0 | **1** | 0 | 0 | 0 | 0 | 0 | 0 | 0 | 0 | 1 |
| $C_5$ | 0 | 0 | **0** | 0 | 0 | 0 | 0 | 0 | **0** | 1 | **0** |
| $C_6$ | 0 | 0 | **1** | 0 | 0 | 0 | 0 | 0 | 0 | 1 | 0 |
| $C_7$ | 0 | 0 | 0 | **0** | 0 | 0 | 0 | 0 | **0** | 1 | 1 |
| $C_8$ | 0 | 0 | 0 | **1** | 0 | 0 | 0 | 0 | 0 | 1 | 1 |
| $C_9$ | 0 | 0 | 0 | 0 | **0** | 0 | 0 | 0 | **1** | **0** | 0 |
| $C_{10}$ | 0 | 0 | 0 | 0 | **1** | 0 | 0 | 0 | 1 | 0 | 0 |
| $C_{11}$ | 0 | 0 | 0 | 0 | 0 | **0** | 0 | 0 | **1** | 0 | **1** |
| $C_{12}$ | 0 | 0 | 0 | 0 | 0 | **1** | 0 | 0 | 1 | 0 | 1 |
| $C_{13}$ | 0 | 0 | 0 | 0 | 0 | 0 | **0** | 0 | **1** | 1 | **0** |
| $C_{14}$ | 0 | 0 | 0 | 0 | 0 | 0 | **1** | 0 | 1 | 1 | 0 |
| $C_{15}$ | 0 | 0 | 0 | 0 | 0 | 0 | 0 | **0** | **1** | **1** | 1 |
| $C_{16}$ | 0 | 0 | 0 | 0 | 0 | 0 | 0 | **1** | 1 | 1 | 1 |

Table 1: The concept class $\mathcal{C}_u^{pair}$ [Zilles et al., 2011], for the case $u = 3$. The subset teaching sets witnessing $\text{STD}(\mathcal{C}_3^{pair}) = 1$ are highlighted in blue. Non-clashing sets that witness $\text{NCTD}(\mathcal{C}_3^{pair}) \leq 2$ are in bold font. The proof of Proposition 17 shows that $\text{NCTD}(\mathcal{C}_3^{pair}) = 2$.

## 3 Desirable Properties of Teaching Models

Our goal is to find a "reasonable" batch teaching complexity measure that is upper-bounded by VCD. Even an upper bound linear in VCD would be of significance; the best known upper bound on batch teaching complexity, in terms of VCD, is the quadratic upper bound on RTD (and thus on NCTD) that was established by Hu et al. [2017]. One important aspect in the design of "reasonable" teaching models is how much to constrain the information exchange between teacher and learner. In an attempt to prevent unfair collusion between a teacher and a learner, one limits their interaction by certain

constraints. While there is no general definition of what constitutes collusion, Goldman and Mathias [1996] proposed a notion of collusion-freeness that is often adopted in the literature:

**Definition 5 (Goldman and Mathias [1996])** *A teacher $T$ and a learner $L$ are Goldman-Mathias (GM) collusion-free on a concept class $\mathcal{C}$ if, for every $C \in \mathcal{C}$, and every sample set $S$ for $C$, we have $L(T(C) \cup S) = C$.*

The intuition behind this notion is that the learner is not distracted when presented with additional information on the target concept, beyond the minimum information needed to identify the target concept. At the same time, this prevents certain "unfair coding tricks". For every concept class $\mathcal{C}$, all TD-teachers, RTD-teachers, and NCTD-teachers can be matched with a learner in a GM-collusion-free way [Zilles et al., 2011, Kirkpatrick et al., 2019]. In particular, $\mathrm{NCTD}(\mathcal{C})$ corresponds to the *smallest* order of any teacher for $\mathcal{C}$ that can be matched with a learner without GM-collusion [Kirkpatrick et al., 2019]. By contrast, teachers and learners using subset teaching usually are not in general GM-collusion-free [Zilles et al., 2011].

As long as it is open whether NCTD is upper-bounded by (a function linear in) VCD, any quest for a measure of batch teaching complexity that is upper-bounded by (a function linear in) VCD is forced to take a new route: it must consider batch teaching notions that *violate* the definition of GM-collusion-freeness, since NCTD is the optimal complexity of GM-collusion-free teaching.

Ignoring GM-collusion-freeness has some practical motivation. The notion of GM-collusion-freeness is inherently limited in that it excludes learners that expect teachers to use only "important" information (for some notion of importance). If $T(C)$ contains only "important" examples for $C$, while the sample $S$ for $C$ contains "unimportant" examples, then a learner may be confused by the presence of $S$ in its input $T(C) \cup S$. If we want to permit superfluous information to have adverse effects in teaching, then we should not require GM-collusion-freeness. As an illustration of the naturalness of such effects, consider a typical form of information exchange between a human teacher and learner in a classroom. Suppose a teacher asks a student to prove Theorem X under premises A, B, and C, and suppose further the student comes up with a proof of Theorem X that uses only premises A and B, but not the (consistent yet unnecessary) premise C. The student may then doubt their own proof, since they assume all the information provided by the teacher to be "important". This is in line with recent research on machine teaching. Ferri et al. [2022] argued that it is *not in general* unnatural for a learner to change its mind when receiving data that is consistent with the previous hypothesis. Their model allows additional information to decrease the probability that the current hypothesis is really correct, and to make an alternate consistent hypothesis appear more likely. This point of view challenges the widely-held belief that natural teaching and learning should be GM-collusion-free.

Consider for example the class in Table 1. Here, $C_1$ and $C_2$ differ only in $x_1$. The smallest teaching set for $C_1$ uses four instances ($x_1$, $x_9$, $x_{10}$, and $x_{11}$), but a learner expecting the teacher to use "important" examples might infer $C_1$ from the information $(x_1, 0)$: while there are 15 concepts consistent with this example, $C_1$ is the only one for which this information is "important", as it is the only concept in the class for which this example occurs in a smallest teaching set. An STD-teacher will use the singleton set $\{(x_1, 0)\}$ to teach $C_1$, and has an order of 1 on this concept class. By contrast, no GM-collusion-free teacher/learner pair can succeed with just a single example, since the NCTD, which is known to be optimal in terms of GM-collusion-free teaching, is greater than 1.

Rather than defining a weaker notion of collusion-freeness, we propose to consider desirable properties of teachers and learners, which could be used as postulates for reasonable models of teaching and learning. Depending on the application setting, one may require additional properties.

Two of our intuitive postulates concern monotonicity of a teaching complexity measure. First, for every concept class $\mathcal{C}$ one might want every superset of $\mathcal{C}$ to be at least as hard to teach as $\mathcal{C}$, i.e., the teaching complexity of a concept class should be no smaller than that of any of its subclasses. Second, one might require that removing instances from $X$ without reducing the number of concepts should not decrease teaching complexity, because the removed instances may have been helpful but not harmful in teaching. Both these monotonicity postulates are captured in the following definition.

**Definition 6 (Postulate 1 (Class-Monotonicity) and Postulate 2 (Domain-Monotonicity))** *Let $Z$ be any function that assigns a non-negative integer to any concept class. We say that $Z$ is class-monotonic, if $Z(\mathcal{C}') \geq Z(\mathcal{C})$ whenever $\mathcal{C}, \mathcal{C}'$ are concept classes over the same domain $X$, and*

$\mathcal{C} \subseteq \mathcal{C}'$. We say that $Z$ is *domain-monotonic*, if $Z(\mathcal{C}\downarrow_{X'}) \geq Z(\mathcal{C})$ whenever $\mathcal{C}$ is a concept class over a domain $X$, and $X' \subseteq X$ is a subset that preserves $\mathcal{C}$.

Our third postulate at first glance appears similar to GM-collusion-freeness. It states that a teaching set for a concept $C$ should not be contained in a teaching set for another concept $C'$. Intuitively, a teaching set $T(C)$ for $C$ must contain enough "important" examples for $C$ such that any concept $C' \neq C$ that is consistent with $T(C)$ will be missing at least one of these "important" examples in its own teaching set. In other words, for every $C' \neq C$ at least one example in the teaching set for $C$ is "unimportant" for $C'$ or inconsistent with $C'$.

**Definition 7 (Postulate 3 (Antichain Property))** *A teacher $T$ has the antichain property (is an antichain teacher) for a concept class $\mathcal{C}$ if $C, C' \in \mathcal{C}$ and $C \neq C' \Rightarrow T(C) \nsubseteq T(C')$.*

We will sometimes say that a teaching complexity notion satisfies the antichain property if all teachers following the corresponding model are antichain teachers. For example, since every TD-teacher is an antichain teacher, we say that TD satisfies the antichain property. It is easy to see that also STD satisfies the antichain property. Moreover, every RTD-teacher (NCTD-teacher) can be normalized to an RTD-teacher (NCTD-teacher, resp.) of the same order that also satisfies the antichain property; see the supplementary material for a proof.

**Proposition 8** *Let $\mathcal{C}$ be any concept class, $Z \in \{\text{RTD}, \text{NCTD}\}$, and $T$ any $Z$-teacher for $\mathcal{C}$. Then there is a $Z$-teacher $T'$ for $\mathcal{C}$ with $\text{ord}(T') = \text{ord}(T)$ such that $T'$ has the antichain property.*

An interesting natural (yet non-trivial) property of optimal antichain teachers is that their order for the powerset $\mathcal{P}_n$ on $n$ instances is linear in $n$. (The proof, given in the supplementary material, relies on the theory of bipartite matching.)

**Theorem 9** *Let $T^n$ be an antichain teacher for $\mathcal{P}_n$ and suppose $\text{ord}(T^n) \leq \text{ord}(T)$ for all antichain teachers $T$ for $\mathcal{P}_n$. Then, for all but finitely many $n$, we have $0.22 \cdot n < \text{ord}(T^n) < 0.23 \cdot n$.*

When teaching with sequences, the sample complexity for $\mathcal{P}_n$ may be as low as $\Theta(\frac{n}{\log n})$ [Mansouri et al., 2019]; "natural" batch teaching models thus need strictly larger sample sets.

To sum up, our goal is to design a measure of batch teaching complexity that is upper-bounded by (a function linear in) VCD and satisfies the postulates of (i) domain-monotonicity, (ii) class-monotonicity and (iii) the antichain property. Since we need to focus on complexity notions violating GM-collusion-freeness, a first natural candidate is STD. However, STD is not even upper-bounded by a polynomial in VCD [Zilles et al., 2011]. Moreover, while it satisfies Postulate 3, it violates both Postulates 1 and 2. The class in Table 1 witnesses that STD is not class-monotonic: it satisfies STD = 1, while its subclass of odd-numbered concepts has STD = 3. This generalizes as follows:

**Proposition 10 (Zilles et al. [2011])** STD *is not class-monotonic. In particular, for every $n > 1$, there is a concept class $\mathcal{C}$ with a sub-class $\mathcal{C}'$ such that $\text{STD}(\mathcal{C}) = 1$, while $\text{STD}(\mathcal{C}') = n$.*

**Proposition 11** STD *is not domain-monotonic. In particular, for every $n > 3$, there is a concept class $\mathcal{C}$ over a domain $X = X' \cup X''$ such that $\text{STD}(\mathcal{C}) = n - 1$, while $\text{STD}(\mathcal{C}\downarrow_{X'}) = 2$.*

## 4 A Variant of STD

Our main contribution is to introduce a new teaching complexity parameter called $\text{STD}_{\min}$, which is built on the idea of STD, yet satisfies the two postulates defined in Section 3. Moreover, we will show that it is upper-bounded by VCD. This makes $\text{STD}_{\min}$ the first known complexity parameter for batch teaching that is proven to be upper-bounded by (a function linear in) VCD.

In calculating $\text{STD}(C, \mathcal{C})$, at stage $k + 1$ one maintains the collection of *all minimum-size* subsets of sets for $C$ from stage $k$ that are not subsets of sets for any other $C'$ at stage $k$. In $\text{STD}_{\min}$, at every stage we pick only one subset, but do not require it to be minimal.

**Definition 12** *Let $\mathcal{C}$ be a concept class over a domain $X$. A sequence $\mathcal{T} = (T_k)_{k \in \mathbb{N}}$ of teachers for $\mathcal{C}$ is called a subset teaching sequence for $\mathcal{C}$ if, for all $C \in \mathcal{C}$ and all $k \in \mathbb{N}$:*

$$
\begin{aligned}
T_0(C) &= \{(x, C(x)) \mid x \in X\}, \\
T_{k+1}(C) &\subseteq T_k(C), \\
T_{k+1}(C) &\not\subseteq T_k(C') \text{ for all } C' \in \mathcal{C},\ C' \neq C.
\end{aligned}
$$

*Let $k^* \in \mathbb{N}$ be minimal with $T_k(C) = T_{k^*}(C)$ for all $k > k^*$ and all $C \in \mathcal{C}$. Then $T_{k^*}$ is an $\mathrm{STD}_{\min}$-teacher for $\mathcal{C}$ and we define the order of $\mathcal{T}$ on $\mathcal{C}$ as $\mathrm{ord}(\mathcal{T}) = \mathrm{ord}(T_{k^*})$. Finally, define*

$$\mathrm{STD}_{\min}(\mathcal{C}) = \min\{\mathrm{ord}(\mathcal{T}) \mid \mathcal{T} \text{ is a subset teaching sequence for } \mathcal{C}\}.$$

**Definition 13** *Let $\mathcal{C}$ be a concept class over a domain $X$. A sequence $\mathcal{T} = (T_k)_{k \in \mathbb{N}}$ of teachers for $\mathcal{C}$ is called a subset teaching sequence for $\mathcal{C}$ if, for all $C \in \mathcal{C}$ and all $k \in \mathbb{N}$:*

$$
\begin{aligned}
T_0(C) &= \{(x, C(x)) \mid x \in X\}, \\
T_{k+1}(C) &\subseteq T_k(C), \\
T_{k+1}(C) &\not\subseteq T_k(C') \text{ for all } C' \in \mathcal{C},\ C' \neq C.
\end{aligned}
$$

*Let $k^* \in \mathbb{N}$ be minimal with $T_k(C) = T_{k^*}(C)$ for all $k > k^*$ and all $C \in \mathcal{C}$. Then $T_{k^*}$ is an $\mathrm{STD}_{\min}$-teacher for $\mathcal{C}$ and we define the order of $\mathcal{T}$ on $\mathcal{C}$ as $\mathrm{ord}(\mathcal{T}) = \mathrm{ord}(T_{k^*})$. Finally, define*

$$\mathrm{STD}_{\min}(\mathcal{C}) = \min\{\mathrm{ord}(\mathcal{T}) \mid \mathcal{T} \text{ is a subset teaching sequence for } \mathcal{C}\}.$$

**Observation 1** *Every subset teaching sequence of order $d$ can be transformed into a* normalized *sequence $(T_k)_{k \in \mathbb{N}}$ of the same order, where a normalized subset teaching sequence has the property that, for every $k < k^*$ and every $C \in \mathcal{C}$, we have (i) $T_{k+1}$ differs from $T_k$ on exactly one concept, (ii) $|T_{k+1}(C)| \in \{|T_k(C)| - 1, |T_k(C)|\}$, (iii) $|T_k(C)| \geq d$, which implies that $|T_{k^*}(C)| = d$.*

Before showing that $\mathrm{STD}_{\min}$ satisfies our desired properties, we claim that $\mathrm{STD}_{\min}$ is upper-bounded by STD and can be arbitrarily smaller than STD (the proof is in the supplementary material):

**Proposition 14** $\mathrm{STD}_{\min}(\mathcal{C}) \leq \mathrm{STD}(\mathcal{C})$*, and for all $n \in \mathbb{N}$ there is some succinct $\mathcal{C}_n$ such that $\mathrm{STD}_{\min}(\mathcal{C}_n) = 2$ and $\mathrm{STD}(\mathcal{C}) = n$.*

Since $\mathrm{STD}_{\min} \leq \mathrm{STD}$ and STD can be smaller than NCTD, which is the best possible complexity of GM-collusion-free teaching and learning, we immediately obtain:

**Corollary 15** *There are concept classes for which no teacher of order $\mathrm{STD}_{\min}$ can be matched with a learner in a GM-collusion-free way.*

Next, we claim that $\mathrm{STD}_{\min}$ also meets Postulates 1 and 2, and thus greatly improves on STD in terms of the postulates defined in Section 3; see the supplementary material for a proof.

**Proposition 16** $\mathrm{STD}_{\min}$ *is class-monotonic, domain-monotonic, and satisfies the antichain property.*

# 5 Relationship of $\mathrm{STD}_{\min}$ to Other Complexity Parameters

We already know that $\mathrm{STD}_{\min}$ is upper-bounded by TD and STD. Subsequently, we will show that, for finite concept classes, it is also upper-bounded by RTD and by VCD, but not by NCTD. Most importantly, our results make $\mathrm{STD}_{\min}$ the first known parameter for batch teaching complexity that is known to be upper-bounded by VCD (or even by a function linear in VCD), for finite classes. We first show that $\mathrm{STD}_{\min}$ can be arbitrarily smaller than NCTD, but also by a factor of 2 larger than NCTD. The proof, which is detailed in Appendix **??**, uses a generalization of $\mathcal{C}_3^{pair}$ from Table 1:

**Proposition 17** *For every $n \in \mathbb{N}$ there is (i) a concept class $\mathcal{C}$ with $\mathrm{STD}(\mathcal{C}) = \mathrm{STD}_{\min}(\mathcal{C}) = 1$ and $\mathrm{NCTD}(\mathcal{C}) = n$; (ii) a concept class $\mathcal{C}$ with $\mathrm{STD}(\mathcal{C}) = \mathrm{STD}_{\min}(\mathcal{C}) = n$ and $\mathrm{NCTD}(\mathcal{C}) = \frac{n}{2}$.*

**Theorem 18** *Let $\mathcal{C}$ be any concept class. Then $\mathrm{STD}_{\min}(\mathcal{C}) \leq \mathrm{RTD}(\mathcal{C})$.*

**Proof.** Let $((C_1, T(C_1)), (C_2, T(C_2)), \ldots, (C_n, T(C_n)))$ be a recursive teaching plan for $\mathcal{C}$. In particular, we have $\mathcal{C} = \{C_1, \ldots, C_n\}$, $|\mathcal{C}| = n$, and, for all $i \in \{1, \ldots, n\}$, the set $T(C_i)$ is a teaching set for $C_i$ with respect to $\{C_j \mid j \geq i\}$. Moreover, $\max_i |T(C_i)| = \mathrm{RTD}(\mathcal{C})$. We will use this recursive teaching plan to construct a subset teaching sequence of order at most $\mathrm{RTD}(\mathcal{C})$ for $\mathcal{C}$.

First, for all $C \in \mathcal{C}$, define $T_0(C)$ as per Definition 13. Second, define $T_1$ by setting $T_1(C_1) = T(C_1)$ and $T_1(C_j) = T_0(C_j)$ for all $j > 1$. Since $T(C_1)$ is a teaching set for $C_1$ wrt $\{C_j \mid j \geq 1\}$, we have that $T_1(C_1) \not\subseteq T_0(C_j)$ for all $j \neq 1$, so that $T_1$ satisfies the requirements of Definition 13. Moreover, $|T_1(C_1)| \leq \mathrm{RTD}(\mathcal{C})$. Next, we define $T_k$ for $k > 1$ by the following algorithm.

1. For $j \neq k$, let $T_k(C_j) = T_{k-1}(C_j)$.
2. To define $T_k(C_k)$, initialize $T_k(C_k) := T(C_k)$.
3. Let $J_k := \{j \neq k \mid T_k(C_k) \subseteq T_{k-1}(C_j)\}$.
4. If $J_k = \emptyset$, then stop and output $T_k(C_k)$.
5. Let $j^* := \min J_k$. Pick any $(x, l) \in T_{k-1}(C_{j^*})$ such that $C_k(x) \neq l$.
6. Let $T_k(C_k) := T_k(C_k) \cup \{(x, C_k(x))\}$.
7. Goto step 3.

Obviously, $T_k(C_k) \subseteq T_{k-1}(C_k)$. To complete the proof of Theorem 18, we show that $T_k(C_k) \not\subseteq T_{k-1}(C)$ for all $C \neq C_k$, $C \in \mathcal{C}$ and that $|T_k(C_k)| \leq \mathrm{RTD}(\mathcal{C})$.

Note that $|T(C_j)| \leq \mathrm{RTD}(\mathcal{C})$ for all $j$; thus $|T_1(C_1)| \leq \mathrm{RTD}(\mathcal{C})$. If $J_k$ is initially empty, then also $|T_k(C_k)| \leq \mathrm{RTD}(\mathcal{C})$, and, by definition of $J_k$ we have $T_k(C_k) \not\subseteq T_{k-1}(C)$ for all $C \neq C_k$. So suppose $J_k$ is not initially empty. Since $T_k(C_k) \supseteq T(C_k)$ at any point in time during the construction, and $T(C_k)$ is a teaching set for $C_k$ wrt $\{C_j \mid j \geq k\}$, we have $j < k$ for all $j \in J_k$ at any point in time. Now let $j^* := \min J_k$. By construction, $T_{k-1}(C_{j^*}) \supseteq T(C_{j^*})$; in particular, $T_{k-1}(C_{j^*})$ is a teaching set for $C_{j^*}$ wrt $\{C_j \mid j \geq j^*\}$. Hence there exists some $(x, l) \in T_{k-1}(C_{j^*})$ such that $C_k(x) \neq l$. Thus, step 6 guarantees that $T_k(C_k) \not\subseteq T_{k-1}(C_{j^*})$, and at least $j^*$ is removed from $J_k$ at the next iteration of step 3. This way, eventually $T_k(C_k) \not\subseteq T_{k-1}(C)$ for all $C \neq C_k$.

All that remains to be shown is that $|T_k(C_k)| \leq \mathrm{RTD}(\mathcal{C})$. We do this by proving inductively for all $k$ that $|T_k(C_j)| \leq \mathrm{RTD}(\mathcal{C})$ for all $j \leq k$. Note that the latter holds true for $k = 1$. Assuming we have proven $|T_{k-1}(C_j)| \leq \mathrm{RTD}(\mathcal{C})$ for all $j \leq k - 1$, we claim that step 6 will only add an element to $T_k(C_k)$ if $|T_k(C_k)| \leq \mathrm{RTD}(\mathcal{C}) - 1$. For step 6 to add an element to $T_k(C_k)$, there must be (at that point in time) a $j^* < k$ such that $T_k(C_k) \subseteq T_{k-1}(C_{j^*})$. Since $T_{k-1}(C_{j^*})$ is a teaching set for $C_{j^*}$ wrt $\{C_j \mid j \geq j^*\}$, the set $T_{k-1}(C_{j^*})$ is inconsistent with $C_k$. In particular, $T_k(C_k)$ is a proper subset of $T_{k-1}(C_{j^*})$. By induction hypothesis, $|T_{k-1}(C_{j^*})| \leq \mathrm{RTD}(\mathcal{C})$. Hence, $|T_k(C_k)| \leq \mathrm{RTD}(\mathcal{C}) - 1$ before step 6 adds an element to $T_k(C_k)$.

In the sum, we have verified that $(T_k)_{k \in \mathbb{N}}$ is a subset teaching sequence for $\mathcal{C}$ and that its order is at most $\mathrm{RTD}(\mathcal{C})$. Thus $\mathrm{STD}_{\min}(\mathcal{C}) \leq \mathrm{RTD}(\mathcal{C})$. $\qquad\square$

The main result of this section establishes $\mathrm{STD}_{\min}$ as the first notion of batch teaching complexity for which VCD is provably an upper bound.

Our proof of this result relies on the recursive decomposition of a concept class, as exploited by [Floyd and Warmuth, 1995] and in the proof that VCD can upper-bound the sample complexity of sequential teaching [Mansouri et al., 2019]. If $\mathcal{C}$ is a concept class over $X$ and $x \in X$ any element of the domain, then we use $\mathcal{C}_x$ to denote $\mathcal{C}\!\downarrow_{X \setminus \{x\}}$, and $\mathcal{C}^x$ to denote $\{C \in \mathcal{C}_x \mid C \cup \{x\} \in \mathcal{C} \text{ and } C \in \mathcal{C}\}$. Floyd and Warmuth call $\mathcal{C}_x$ the restriction and $\mathcal{C}^x$ the reduction of $\mathcal{C}$ wrt $x$. It is easy to see that $\mathrm{VCD}(\mathcal{C}_x) \leq \mathrm{VCD}(\mathcal{C})$ and $\mathrm{VCD}(\mathcal{C}^x) \leq \mathrm{VCD}(\mathcal{C}) - 1$ [Floyd and Warmuth, 1995].

**Theorem 19** *Let $\mathcal{C}$ be any concept class. Then $\mathrm{STD}_{\min}(\mathcal{C}) \leq \mathrm{VCD}(\mathcal{C})$.*

**Proof.** Let $d = \mathrm{VCD}(\mathcal{C}) \geq 1$. Pick some $x^* \in X$. We may assume inductively that the assertion of the theorem is true for the classes $\mathcal{C}_{x^*}$ and $\mathcal{C}^{x^*}$. Let $\mathcal{T}^{\mathrm{rest}} = (T_k^{\mathrm{rest}})_{k \leq k_{\mathrm{rest}}}$ be any subset teaching sequence for $\mathcal{C}_{x^*}$ with $\mathrm{ord}(\mathcal{T}^{\mathrm{rest}}) = d$. Here $k_{\mathrm{rest}}$ denotes the maximum number of iterations $\mathcal{T}^{\mathrm{rest}}$ takes to converge on any concept in $\mathcal{C}_{x^*}$. Further, let $\mathcal{T}^{\mathrm{red}} = (T_k^{\mathrm{red}})_{k \leq k_{\mathrm{red}}}$ be any subset teaching sequence for $\mathcal{C}^{x^*}$ with $\mathrm{ord}(\mathcal{T}^{\mathrm{red}}) = d - 1$. Here $k_{\mathrm{red}}$ denotes the maximum number of iterations $\mathcal{T}^{\mathrm{red}}$ takes to converge. Since any subset teaching sequence can be normalized, it can be assumed in the sequel that the sequences $\mathcal{T}^{\mathrm{rest}}$ and $\mathcal{T}^{\mathrm{red}}$ have the normalization properties (i)—(iii) as in Observation 1. Now define a subset teaching sequence $\mathcal{T} = (T_k)_{k \in \mathbb{N}}$ for $\mathcal{C}$ as follows.

For all $C \in \mathcal{C}$ initialize $T_0(C) := \{(x, C(x)) \mid x \in X\}$. From now on, we use $\mathcal{C}^{x^*,X}$ to denote the set of concepts $C \in \mathcal{C}$ for which there is some $C' \in \mathcal{C}$ such that $C$ and $C'$ differ only in the value of $x^*$. (Note that $\mathcal{C}^{x^*,X}$ has twice as many concepts as $\mathcal{C}^{x^*}$ and its restriction wrt $x^*$ yields $\mathcal{C}^{x^*}$.)

*Stage 1.* Let $k \in [1, k_{\text{red}} + 1]$. For all $C \in \mathcal{C}^{x^*,X}$ set $T_k(C) := T_{k-1}^{\text{red}}(C{\downarrow}_{X \setminus \{x^*\}}) \cup \{(x^*, C(x^*))\}$. For all $C \in \mathcal{C} \setminus \mathcal{C}^{x^*,X}$ set $T_k(C) := \{(x, C(x)) \mid x \in X \setminus \{x^*\}\}$.[1]

*Stage 2.* Let $k \in [k_{\text{red}} + 2, k_{\text{red}} + k_{\text{rest}} + 1]$. For all $C \in \mathcal{C}^{x^*,X}$, let $T_k(C) := T_{k-1}(C)$. For all $C \in \mathcal{C} \setminus \mathcal{C}^{x^*,X}$, let $T_k(C) := T_{k-k_{\text{red}}-1}^{\text{rest}}(C{\downarrow}_{X \setminus \{x^*\}})$.

We claim that $(T_k)_{k \leq k_{\text{red}} + k_{\text{rest}} + 1}$ is a subset teaching sequence for $\mathcal{C}$ and has order at most $d$.

First, we show that the partial sequence defined in Stage 1 is valid, i.e., for all $k \leq k_{\text{red}} + 1$ and all $C, C' \in \mathcal{C}$ with $C \neq C'$ we have $T_k(C) \subseteq T_{k-1}(C)$ and $T_k(C) \not\subseteq T_{k-1}(C')$. The fact $T_k(C) \subseteq T_{k-1}(C)$ is inherited from the same fact for the sequence $\mathcal{T}^{\text{red}}$. To show that $T_k(C) \not\subseteq T_{k-1}(C')$, consider multiple cases. If $C, C' \in \mathcal{C}^{x^*,X}$, this property is again inherited from $\mathcal{T}^{\text{red}}$. If $C \in \mathcal{C}^{x^*,X}$ and $C' \notin \mathcal{C}^{x^*,X}$, then $(x^*, C(x^*)) \in T_k(C) \setminus T_{k-1}(C')$ when $k > 1$, while for $k = 1$ the property $T_k(C) \not\subseteq T_{k-1}(C')$ is valid because $T_1(C) = T_0(C)$. If $C \notin \mathcal{C}^{x^*,X}$, then $C$ and $C'$ must differ on some instance $x \neq x^*$; since $T_k(C)$ contains $(x, C(x))$, again we obtain $T_k(C) \not\subseteq T_{k-1}(C')$.

Second, we show that the partial sequence defined in Stage 2 is a valid extension of that defined in Stage 1, i.e., for $k_{\text{red}} + 2 \leq k \leq k_{\text{red}} + k_{\text{rest}} + 1$ and all $C, C' \in \mathcal{C}$ with $C' \neq C$ we have $T_k(C) \subseteq T_{k-1}(C)$ and $T_k(C) \not\subseteq T_{k-1}(C')$. So, fix $C$ and $C' \neq C$. For $C \in \mathcal{C}^{x^*,X}$, the sets defined in Stage 1 do not get updated in Stage 2. Hence we may safely assume that $C \notin \mathcal{C}^{x^*,X}$. $T_k(C) \subseteq T_{k-1}(C)$ is inherited from the same property for $\mathcal{T}^{\text{rest}}$. If $C' \notin \mathcal{C}^{x^*,X}$, then the analogous remark applies to $T_k(C) \not\subseteq T_{k-1}(C')$. Suppose now that $C' \in \mathcal{C}^{x^*,X}$. Here we make use of the normalization properties (i)—(iii). In Stage 2, all sets $T_{k-1}(C')$ with $C' \in \mathcal{C}^{x^*,X}$ are of size $d$ while all sets $T_k(C)$ with $C \notin \mathcal{C}^{x^*,X}$ are of size at least $d$. Since $T_{k-1}(C')$ contains $(x^*, C(x^*))$, it is not a subset of $T_k(C)$. Together with $|T_k(C)| \geq |T_{k-1}(C')|$, it now follows that $T_k(C) \not\subseteq T_{k-1}(C')$. Consequently, $\mathcal{T}$ is a subset teaching sequence for $\mathcal{C}$. The identity $\text{ord}(\mathcal{T}) = d$ is obvious. $\qquad\square$

# 6 A Note on Unambiguity in Teaching

While $\text{STD}_{\min}$ meets our objectives outlined earlier, it loses one potentially desirable property of STD that has not been addressed in the literature before. For every teaching model defined above, there are concept classes which can be taught with more than one teacher satisfying the requirements of the model. Suppose Teacher 1 teaches concept $C_1 \in \mathcal{C}$ using sample set $S$, while Teacher 2 teaches concept $C_2 \in \mathcal{C}$, $C_2 \neq C_1$, with $S$. If the learner knows that she is interacting with *some* teacher of a given model, but doesn't know which one, she then cannot disambiguate between $C_1$ and $C_2$ when given input $S$. In *unambiguous* teaching models, the learner can infer the target concept from the given sample set without being told with which of the multiple valid teachers she is interacting.

**Definition 20** *Let $Z \in \{\text{TD}, \text{RTD}, \text{NCTD}, \text{STD}, \text{STD}_{\min}\}$ be a teaching complexity parameter. We say that $Z$ is unambiguous on a concept class $\mathcal{C}$, if, for every sample set $S$, there is at most one concept $C \in \mathcal{C}$ such that some $Z$-teacher $T$ of order $Z(\mathcal{C})$ satisfies $T(C) = S$.*

If $Z$ is ambiguous on a concept class, then one would have to normalize $Z$-teachers or use disambiguating information. One disadvantage of $\text{STD}_{\min}$ compared to STD is the loss of unambiguity.

**Theorem 21** TD*,* RTD*, and* STD *are unambiguous on every concept class, while there are concept classes on which* NCTD *and* $\text{STD}_{\min}$ *are not unambiguous.*

**Proof.** Let $\mathcal{C}$ be any concept class and $S$ a sample set. If a TD-teacher $T$ maps $C$ to $S$, then $S$ is a teaching set for $C$ and thus cannot be mapped to any other concept by any TD-teacher $T'$. If an RTD-teacher $T$ maps $C$ to $S$ and $C \in \mathcal{C}_i^{\min}$, then $S$ is inconsistent with the concepts in $C \in \mathcal{C}_i \setminus \{C\}$ and thus cannot be mapped to the latter. The concepts in $\mathcal{C} \setminus \mathcal{C}_i$ cannot be mapped to $S$ by any RTD-teacher either, since they must be mapped to sample sets that are inconsistent with all concepts

---

[1]Note that, for all $C \in \mathcal{C}^{x^*,X}$, we have that $T_1(C)$ still equals $T_0(C)$.

in $C \in \mathcal{C}_i$, which includes $C$. If an STD-teacher $T$ maps $C$ to $S$, then $S \in \text{STS}^{k^*}(C, \mathcal{C})$. Since the latter is disjoint from $S \in \text{STS}^{k^*}(C', \mathcal{C})$ for all $C' \neq C$, no STD-teacher can map such $C'$ to $S$.

It is easy to see that there are two NCTD-teachers of order 1 for $\mathcal{P}_2$ that even have the same range, one maps $\emptyset$ to $\{(x_1, 0)\}$ and $\{x_1\}$ to $\{(x_2, 0)\}$, the other one does the opposite. A concept class on which $\text{STD}_{\min}$ is ambiguous is Warmuth's class [Doliwa et al., 2014], which is known as the smallest class on which RTD exceeds VCD. For details, we refer to the supplementary material. $\square$

The ambiguity observed for $Z \in \{\text{NCTD}, \text{STD}_{\min}\}$ is stronger than the negation of the condition in Definition 20. Ambiguity means that multiple optimal $Z$-teachers map two different concepts to the same teaching set. But it even happens that two optimal $Z$-teachers $T_1 \neq T_2$ have exactly the same range $R = \{T_1(C) \mid C \in \mathcal{C}\} = \{T_2(C) \mid C \in \mathcal{C}\}$, while assigning the sample sets in $R$ differently to the concepts in $C$. The ambiguity of $\text{STD}_{\min}$ can take extreme forms for artificially created concept classes that have many redundant instances, as discussed in the supplementary material.

## 7 Conclusions

In an abstract setting, devoid of any specific application, it is not straightforward to define what makes a teaching/learning model "natural". We addressed this problem by considering various properties of teaching that one might find desirable; one can then design models that possess (some of) these properties as seen fit in an underlying application setting. We have introduced some such properties, here called "postulates", which are fulfilled by $\text{STD}_{\min}$. Another property of $\text{STD}_{\min}$, that is not fulfilled by TD, RTD or NCTD, is that it allows the learner to discard a hypothesis because the given data is not *important* for that hypothesis (even if it is consistent). We argue that a learner who knows that data comes from a helpful teacher, has reasons to reject hypotheses for which the data is not important/helpful. Of course, there may be other natural properties that are not fulfilled by $\text{STD}_{\min}$. An interesting research question is what natural properties can (or cannot) be fulfilled by teaching models whose parameter is upper-bounded by VCD.

This paper is the first to provide an upper bound of VCD on batch teaching complexity. Our results and proofs may have implications on the design and analysis of teaching and learning algorithms. For an overview of most of our results, see Table 2. While we diverted from the notion of GM-collusion-freeness, we hope that the postulates on teaching that we defined, as well as our notion of unambiguity, will be useful for future studies on machine teaching. In particular, noting that neither NCTD nor $\text{STD}_{\min}$ satisfy the notion of unambiguity, one may ask the question whether ambiguity is inherent in "natural" teaching models that yield a complexity upper-bounded by VCD. All unambiguous models studied in this paper have a complexity that sometimes exceeds VCD.

| Property | TD | RTD | NCTD | STD | **STD**$_{\min}$ |
|---|---|---|---|---|---|
| GM-collusion-freeness [Goldman and Mathias, 1996] | yes | yes | yes | no | **no** |
| P1 – class-monotonicity | yes | yes | yes | no | **yes** |
| **P2 – domain-monotonicity** | **yes** | **yes** | **yes** | **no** | **yes** |
| **P3 – antichain property** | **yes** | **yes*** | **yes*** | **yes** | **yes** |
| **unambiguity** | **yes** | **yes** | **no** | **yes** | **no** |
| $\leq$ RTD | no | yes | yes | no | **yes** |
| $\leq$ VCD | no | no | ? | no | **yes** |
| $\leq$ polynomial in VCD | no | yes | yes | no | **yes** |

Table 2: Comparison of teaching complexity notions. Contributions from this paper are highlighted in bold. Note that class-monotonicity was already discussed by Zilles et al. [2011]. The asterisks indicate that RTD- and NCTD-teachers can be normalized to antichain teachers, see Prop. 8.

**Acknowledgments and Disclosure of Funding**

This research was conducted while S. Zilles was a visiting scientist at the Max Planck Institute for Software Systems. Further, S. Zilles acknowledges financial support by the Natural Sciences and Engineering Research Council (NSERC) of Canada, both through the Canada Research Chairs program and the Discovery Grants program. She was also supported through the Canada CIFAR AI Chairs program as an Affiliate Chair with the Alberta Machine Intelligence Institute (Amii).

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
