# Supplementary Material for Paper 6775
# (On Batch Teaching with Sample Complexity Bounded by VCD)

## Abstract

This paper contains proof details omitted from the main paper as well as a more detailed discussion of the ambiguity of $\mathrm{STD}_{\min}$-teaching.

## A   Proof of Theorem 9

**Theorem 9** *Let $T^n$ be an antichain teacher for $\mathcal{P}_n$ and suppose $\mathrm{ord}(T^n) \leq \mathrm{ord}(T)$ for all antichain teachers $T$ for $\mathcal{P}_n$. Then, for all but finitely many $n$, we have $0.22 \cdot n < \mathrm{ord}(T^n) < 0.23 \cdot n$.*

To establish this result, we first introduce some notation and some background on bipartite matching.

**Definition 21** *Let $\mathcal{C}$ be any concept class. The antichain number of $\mathcal{C}$, denoted by $\mathrm{ACN}(\mathcal{C})$, is the smallest possible order of a teacher for $\mathcal{C}$ with the antichain property.*

Theorem 9 can then be restated as follows:

*For all but finitely many $n$, we have $0.22 \cdot n < \mathrm{ACN}(\mathcal{P}_n) < 0.23 \cdot n$.*

It is well known that a bipartite graph all of whose vertices have the same degree contains a perfect matching. The simple proof is based on a double counting argument. The same kind of argument can be used to show the following (most likely also well known) result:

**Lemma 22** *Let $G = (V_1, V_2, E)$ be a bipartite graph with vertex sets $V_1$ and $V_2$. Suppose that every vertex in $V_1$ has degree $d_1$ while every vertex in $V_2$ has degree $d_2 \leq d_1$. Then $G$ contains a matching of size $|V_1|$.*

**Proof.** For $U \subseteq V_1$, $\Gamma(U)$ denotes the neighborhood of $U$, i.e., $\Gamma(U) = \{v \in V_1 \mid v$ is adjacent to some vertex in $U\}$. It suffices to show that Hall's condition,

$$\forall U \subseteq V_1 : |\Gamma(U)| \geq |U| \ ,$$

is satisfied. Fix a set $U \subseteq V_1$. The number of edges having one endpoint in $U$ equals $d_1 \cdot |U|$. The number of edges having one endpoint in $\Gamma(U)$ is at most $d_2 \cdot |\Gamma(U)|$. An edge with an endpoint in $U$ must have its other endpoint in $\Gamma(U)$. Hence $d_1 \cdot |U| \leq d_2 \cdot |\Gamma(U)|$. Since $d_2 \leq d_1$, we may conclude that $|U| \leq |\Gamma(U)|$. $\qquad\square$

**Corollary 23** *Let $d, n$ be integers such that $1 \leq d \leq (n+1)/2$. Let $X$ be a set of size $n$. Let $G = (V_1, V_2, E)$ be the bipartite graph such that*

- *$V_1$ (resp. $V_2$) consists of all subsets of $X$ with $d-1$ (resp. $d$) elements,*

26     • *a set $U \in V_1$ is adjacent to a set $U' \in V_2$ iff $U \subseteq U'$.*

27     *Then $G$ contains a a matching of size $|V_1|$.*

28 **Proof.** Each vertex in $V_1$ has degree $n - d + 1$ whereas each vertex in $V_2$ has degree $d$. Since
29 $d \le (n+1)/2$, by assumption, it follows that $d \le n - d + 1$. Now apply Lemma 22.     $\square$

30 Let $X$ be a set of size $n$. A sample set over $X$ is said to be *conflict-free* if it does not contain both
31 $(x,0)$ and $(x,1)$ for some $x \in X$. Let $\mathcal{F}_{\le d,n}$ be the family of all conflict-free sample sets over $X$
32 with $d$ or fewer elements. The conflict-free sample sets with exactly $d$ elements form an antichain –
33 denoted by $\mathcal{F}_{=d,n}$ in the sequel – in $\mathcal{F}_{\le d}$. Obviously

$$\mathcal{F}_{=d,n} = \{(x_1,b_1),\ldots,(x_d,b_d) : x_1,\ldots,x_d \text{ are } d \text{ distinct elements of } X \text{ and } b_1,\ldots,b_d \in \{0,1\}\}$$

34 and therefore the antichain $\mathcal{F}_{=d,n}$ is of size $\binom{n}{d} \cdot 2^d$.

35 The following result is a relative of Sperner's Theorem:

36 **Lemma 24** *$\mathcal{F}_{=d,n}$ is a maximum antichain in $\mathcal{F}_{\le d,n}$.*

37 **Proof.** An antichain $\mathcal{A}'$ with conflict-free sets $A'_1,\ldots,A'_{s'}$ (without repetition) is called an *extension*
38 of another antichain $\mathcal{A}$ with conflict-free sets $A_1,\ldots,A_s$ (again without repetition) if $s' = s$ and
39 $A_i \subseteq A'_i$ for $i = 1,\ldots,s$ (after renumbering the sets in $\mathcal{A}'$ if necessary). We show, by induction on
40 $d$, that every antichain $\mathcal{A}$ with sets taken from $\mathcal{F}_{\le d,n}$ has an extension $\mathcal{A}'$ with sets taken from $\mathcal{F}_{=d,n}$.
41 For $d = 1$, this is obviously true. Let $d \ge 2$ and assume inductively that it holds for $d - 1$. Fix an
42 antichain $\mathcal{A}$ with sets taken from $\mathcal{F}_{\le d}, n$. Let $\mathcal{A}_1$ be the antichain consisting of the sets of size at
43 most $d - 1$ in $\mathcal{A}$ and let $\mathcal{A}_2 = \mathcal{A} \setminus \mathcal{A}'$. By our inductive assumption, there is an extension $\mathcal{A}'_1$ of $\mathcal{A}_1$
44 whose sets are taken from $\mathcal{F}_{\le d-1,n}$. The inductive proof can now be accomplished by proving the
45 following assertions:

46 **Claim 1:** $\mathcal{A}'_1 \cup \mathcal{A}_2$ is an antichain in $\mathcal{F}_{\le d,n}$ whose sets are of size $d - 1$ or $d$.
47 **Claim 2:** Any antichain $\mathcal{B}$ with sets of size $d - 1$ or $d$ has an extension $\mathcal{B}'$ with sets taken from
48        $\mathcal{F}_{\le d,n}$.

49 Claim 1 becomes obvious from the following observations:

50     • No set in $\mathcal{A}_2$ (with $d$ elements) can be a subset of some set in $\mathcal{A}'_1$ (with $d - 1$ elements).
51     • Since no set in $\mathcal{A}_1$ is a subset of some set in $\mathcal{A}_2$ (by the antichain property of $\mathcal{A}$), no set in
52        the extension $\mathcal{A}'_1$ is a subset of some set in $\mathcal{A}_2$.

53 As for proving Claim 2, fix some antichain $\mathcal{B}$. Let $\mathcal{B} = \mathcal{B}_1 \cup \mathcal{B}_2$ be the decomposition of $\mathcal{B}$ into sets of
54 size $d-1$ and sets of size $d$, respectively. A set of $\mathcal{B}_1$ is of the form $B = \{(x_1,b_1),\ldots,(x_{d-1},b_{d-1})\}$.
55 Let $M$ be the matching of size $|V_1|$, whose existence is guaranteed by Corollary 23. Pick $x_d$ such
56 that $\{x_1,\ldots,x_{d-1},x_d\}$ is the $M$-partner of $\{x_1,\ldots,x_{d-1}\}$. Then the set

$$B' = \{(x_1,b_1),\ldots,(x_{d-1},b_{d-1}),(x_d,0)\}$$

57 is called the $M$-partner of $B$. Note here that different sets from $\mathcal{B}_1$ have different $M$-partners.
58 Let $\mathcal{B}'_1$ be the antichain obtained from $\mathcal{B}_1$ by replacing each set $B$ in $\mathcal{B}_1$ by its $M$-partner and let
59 $\mathcal{B}' = \mathcal{B}'_1 \cup \mathcal{B}_2$. By construction, all sets in $\mathcal{B}'$ are of size $d$. In order to show that $\mathcal{B}'$ is an antichain
60 that extends $\mathcal{B}$, it suffices to show that no $M$-partner of a set $B \in \mathcal{B}_1$ can be equal to one of the sets
61 in $\mathcal{B}_2$. But this is obvious because $B$ is a subset of its $M$-partner, but not a subset of any set in $\mathcal{B}_2$ (by
62 the antichain property of $\mathcal{B}$). Claim 2 follows from this discussion, which also completes the proof of
63 the lemma.     $\square$

64 **Corollary 25** *Let $d_0 = d_0(n)$ be the smallest $d$ such that $2^d \cdot \binom{n}{d} \ge 2^n$. Let $G = (V_1, V_2, E)$ be the*
65 *bipartite graph given by (i) $V_1 = \mathcal{F}_{=n,n}$ and $V_2 = \mathcal{F}_{=d_0,n}$, and (ii) a set $U' \in V_1$ is adjacent to a set*
66 *$U \in V_2$ iff $U \subseteq U'$. Then $G$ contains a matching of size $|V_1|$.*

67 **Proof.** Each vertex in $V_1$ has degree $\binom{n}{d_0}$ whereas each vertex in $V_2$ has degree $2^{n-d_0}$. The definition
68 of $d_0$ implies that $2^{n-d_0} \le \binom{n}{d_0}$. Now apply Lemma 22.     $\square$

69 Note that $\mathrm{ACN}(\mathcal{C})$ is upper-bounded by the smallest number $d$ such that the following graph $G =$
70 $(V_1, V_2, E)$ contains a matching $M$ that matches every vertex in $V_1$: (i) $V_1 = \mathcal{C}$ and $V_2 = \mathcal{F}_{=d,n}$, (ii)
71 a concept $C \in \mathcal{C}$ is adjacent to a sample $S \in \mathcal{F}_{=d,n}$ iff it is consistent with $S$.

72 We now obtain a non-trivial reformulation of ACN:

**Theorem 26** *Let* $|X| = n$ *and let* $d_0 = d_0(n)$ *be the smallest* $d$ *such that* $2^d \cdot \binom{n}{d} \geq 2^n$. *Then*
74 $\mathrm{ACN}(\mathcal{P}_n) = d_0(n)$.

**Proof.** Note that $\mathcal{P}_n$ can be identified with $\mathcal{F}_{=n,n}$: each map $C : X \to \{0,1\}$ is identified with the
76 full sample $\{(x, C(x)) \mid x \in X\}$. An application of Corollary 25 yields $\mathrm{ACN}(\mathcal{P}_n) \leq d_0(n)$.

77 Set $d = \mathrm{ACN}(\mathcal{P}_n)$. Then the maximum antichain in $\mathcal{F}_{\leq d,n}$ is of size at least $|\mathcal{P}_n| = 2^n$. Using
78 Lemma 24 and the fact that $|\mathcal{F}_{=d,n}| = \binom{n}{d} \cdot 2^d$, this translates into $2^d \cdot \binom{n}{d} \geq 2^n$. The definition of
79 $d_0(n)$ now implies that $d \geq d_0(n)$. $\qquad\square$

80 We now show that $d_0(n)$ is a function linear in $n$.

**Lemma 27** *Let* $d_0 = d_0(n)$ *be the smallest* $d$ *such that* $2^d \cdot \binom{n}{d} \geq 2^n$. *Then* $0.22 \cdot n < d_0(n) < 0.23 \cdot n$
82 *for all but finitely many* $n$.

**Proof.** For $d = n/2$, we have $\binom{n}{n/2} \asymp \sqrt{\frac{2}{\pi n}} 2^n$, which is asymptotically larger than $2^{n/2}$. We
84 may therefore assume that $d \leq n/2$. For such $d$, the term $\binom{n}{d}$ decreases when $d$ decreases, while
85 $2^{n-d}$ increases. Hence it suffices to show that $2^d \cdot \binom{n}{d} \geq 2^n$ is fulfilled for large enough $n$ when
86 $d = 0.23 \cdot n$, while it is not fulfilled for large enough $n$ when $d = 0.22 \cdot n$.

87 To this end, let $d = pn$ with $0 < p \leq 1/2$, and rewrite $2^d \cdot \binom{n}{d} \geq 2^n$ as

$$\frac{1}{n} \log \binom{n}{pn} \geq 1 - p.$$

88 It is well known that the left-hand side converges to $H(p)$, where $H(\cdot)$ denotes the binary entropy.
89 The lemma now follows from $H(0.22) < 0.78 = 1 - 0.22$ and $H(0.23) > 0.77 = 1 - 0.23$. $\qquad\square$

90 This allows us to conclude that, asymptotically, the value of $\mathrm{ACN}(\mathcal{P}_n)$ lies between $0.22 \cdot n$ and
91 $0.23 \cdot n$, as claimed by Theorem 9.

# B   Other Proof Details for Section 3

**Proposition 8** *Let* $\mathcal{C}$ *be any concept class,* $Z \in \{\mathrm{RTD}, \mathrm{NCTD}\}$, *and* $T$ *any* $Z$-teacher for $\mathcal{C}$. *Then*
94 *there is a* $Z$-teacher $T'$ for $\mathcal{C}$ with $\mathrm{ord}(T') = \mathrm{ord}(T)$ *such that* $T'$ *has the antichain property.*

**Proof.** First, let $T$ be any NCTD-teacher for $\mathcal{C}$. For $C \in \mathcal{C}$, obtain $T'(C)$ from $T(C)$ as follows. If
96 each sample set in $T(C)$ has size $\mathrm{ord}(T)$, then $T'(C) = T(C)$. Otherwise, $T'(C)$ results from $T(C)$
97 by adding examples that are consistent with $C$ to every sample set $T_C \in T(C)$, until the size of $T_C$
98 equals $\mathrm{ord}(T)$. Then $T'$ inherits the non-clashing property on $\mathcal{C}$ from $T$. Clearly, a non-clashing
99 teacher mapping that produces only sample sets of a constant size must also fulfill the antichain
100 property. So $T'$ is an NCTD-teacher for $\mathcal{C}$ with the antichain property, and $\mathrm{ord}(T') = \mathrm{ord}(T)$.

101 Second, suppose $T$ is an RTD-teacher. The construction of $T'$ is identical to that in the first case.
102 It remains to verify that the resulting antichain teacher $T'$ with $\mathrm{ord}(T') = \mathrm{ord}(T)$ is also an RTD-
103 teacher for $\mathcal{C}$. Using the notation from Definition 2, we know that, for $C \in \mathcal{C}_i^{\min}$, the set $T(C)$ is
104 a teaching set for $C$ wrt $\mathcal{C}_i$. Since adding examples (consistently with $C$) to $T(C)$ does not change
105 this fact, we obtain that, for $C \in \mathcal{C}_i^{\min}$, the set $T'(C)$ is a teaching set for $C$ wrt $\mathcal{C}_i$. Hence $T'$ is an
106 RTD-teacher for $\mathcal{C}$. $\qquad\square$

**Proposition 11** STD *is not domain-monotonic. In particular, for every* $n > 3$, *there is a concept*
108 *class* $\mathcal{C}$ *over a domain* $X = X' \cup X''$ *such that* $\mathrm{STD}(\mathcal{C}) = n - 1$, *while* $\mathrm{STD}(\mathcal{C}{\downarrow}_{X'}) = 2$.

**Proof.** Let $n > 3$, and let $X' = \{x_1', \ldots, x_n'\}$ and $X'' = \{x_1'', \ldots, x_n''\}$. For every $J \subseteq [n]$ of size 1
110 or 2, let $C_J$ be the concept that assigns label 1 (resp. label 0) to $x_j'$ and $x_j''$ if $j \in J$ (resp. if $j \notin J$).
111 Let $C_\emptyset$ be the concept that assigns label 0 to $x_1', \ldots, x_n'$ and label 1 to $x_1'', \ldots, x_n''$. Consider now the

following concept class $\mathcal{C}$ over the domain $X = X' \cup X''$: $\mathcal{C} = \{C_J \mid J \subseteq [n],\ 0 \le |J| \le 2\}$. See Table 3 for an illustration of the case $n = 5$.

Note that $\mathcal{C}{\downarrow_{X'}}$ is the class of all subsets of $X$ whose size is at most 2. It is well known [Zilles et al., 2011] that $\mathrm{STD}(\mathcal{C}{\downarrow_{X'}}) = 2$.

It remains to prove that $\mathrm{STD}(\mathcal{C}) = n - 1$. To this end, we first determine the minimum teaching sets for every concept in $\mathcal{C}$:

(i) The minimum teaching sets for $C_\emptyset$ are the sets of the form $\{(x'_j, 0), (x''_j, 1)\}$ for $j = 1, \ldots, n$.

(ii) For $1 \le i < j \le n$, the minimum teaching sets for $C_{\{i,j\}}$ are the sets of the form $\{(u_i, 1), (u_j, 1)\}$ where $u_i \in \{x'_i, x''_i\}$, $u_j \in \{x'_j, x''_j\}$ and $\{u_i, u_j\} \cap \{x'_i, x'_j\} \ne \emptyset$.

(iii) For $1 \le i \le n$, the minimum teaching sets for $C_{\{i\}}$ are the sets of the form $\{(u_j, 0) \mid j \in [n] \setminus \{i\}\}$ where $u_j \in \{x'_j, x''_j\}$ and, for at least one index $j' \in [n] \setminus \{i\}$, we have $u_{j'} = x''_{j'}$.

For each $C \in \mathcal{C}$, let $\mathrm{TS}(C)$ be the collection of minimum teaching sets for $C$. The largest of these minimum teaching sets, namely the ones for concepts of the form $C_{\{i\}}$, are of size $n - 1$. Hence $\mathrm{TD}(\mathcal{C}) = n - 1$. Next, we will verify the following property for every concept $C \in \mathcal{C}$:

(*) If $S$ is a minimum teaching set for $C$ wrt $\mathcal{C}$, then every proper subset of $S$ is contained in a minimum teaching set for some concept $C'$ wrt $\mathcal{C}$, where $C' \in \mathcal{C}$, $C' \ne C$.

(i) Consider an index $j \in [n]$ and a teaching set $\{(x'_j, 0), (x''_j, 1)\} \in \mathrm{TS}(C_\emptyset)$. Removing $(x'_j, 0)$ from this set yields a subset of one of the teaching sets for $C_J \ne C_\emptyset$ whenever $j \in J$ and $|J| = 2$. A similar reasoning applies when removing $(x''_j, 1)$ instead of $(x'_j, 0)$.

(ii) Consider indices $i \ne j \in [n]$ and a teaching set $\{(u_i, 1), (u_j, 1)\} \in \mathrm{TS}(C_{\{i,j\}})$. Removing one example, say $(u_i, 1)$, from this set yields a subset of one of the teaching sets for $C_J \ne C_{\{i,j\}}$ whenever $j \in J$, $i \notin J$ and $|J| = 2$.

(iii) Consider an index $i \in [n]$ and a teaching set $\{(u_j, 0) \mid j \in [n] \setminus \{i\}\} \in \mathrm{TS}(C_{\{i\}})$. Removing $(u_{j_0}, 0)$ from this set yields a subset of one of the teaching sets for $C_{\{j_0\}}$.

This establishes Property (*), which immediately implies $\mathrm{STD}(\mathcal{C}) = \mathrm{TD}(\mathcal{C}) = n - 1$. $\qquad\square$

| concept | $x_1$ | $x_2$ | $x_3$ | $x_4$ | $x_5$ | $x'_1$ | $x'_2$ | $x'_3$ | $x'_4$ | $x'_5$ |
|---|---|---|---|---|---|---|---|---|---|---|
| $C_\emptyset$ | 0 | 0 | 0 | 0 | **0** | 1 | 1 | 1 | 1 | **1** |
| $C_{\{1\}}$ | 1 | 0 | 0 | 0 | 0 | 1 | **0** | **0** | **0** | **0** |
| $C_{\{2\}}$ | **0** | 1 | 0 | 0 | 0 | 0 | 1 | **0** | **0** | **0** |
| $C_{\{3\}}$ | **0** | **0** | 1 | 0 | 0 | 0 | 0 | 1 | **0** | **0** |
| $C_{\{4\}}$ | **0** | **0** | **0** | 1 | 0 | 0 | 0 | 0 | 1 | **0** |
| $C_{\{5\}}$ | **0** | **0** | **0** | 0 | 1 | 0 | 0 | 0 | **0** | 1 |
| $C_{\{1,2\}}$ | **1** | **1** | 0 | 0 | 0 | 1 | 1 | 0 | 0 | 0 |
| $C_{\{1,3\}}$ | **1** | 0 | **1** | 0 | 0 | 1 | 0 | 1 | 0 | 0 |
| $C_{\{1,4\}}$ | **1** | 0 | 0 | **1** | 0 | 1 | 0 | 0 | 1 | 0 |
| $C_{\{1,5\}}$ | **1** | 0 | 0 | 0 | **1** | 1 | 0 | 0 | 0 | 1 |
| $C_{\{2,3\}}$ | 0 | **1** | **1** | 0 | 0 | 0 | 1 | 1 | 0 | 0 |
| $C_{\{2,4\}}$ | 0 | **1** | 0 | **1** | 0 | 0 | 1 | 0 | **1** | 0 |
| $C_{\{2,5\}}$ | 0 | **1** | 0 | 0 | **1** | 0 | 1 | 0 | 0 | **1** |
| $C_{\{3,4\}}$ | 0 | 0 | **1** | **1** | 0 | 0 | 0 | **1** | **1** | 0 |
| $C_{\{3,5\}}$ | 0 | 0 | **1** | 0 | **1** | 0 | 0 | **1** | 0 | 1 |
| $C_{\{4,5\}}$ | 0 | 0 | 0 | **1** | **1** | 0 | 0 | 0 | **1** | 1 |

Table 3: The concept class $\mathcal{C}$ from the proof of Proposition 11 for $n = 5$. The entries in bold indicate one (arbitrarily chosen) minimum teaching set for each concept.

## C   Proof Details for Section 4

**Observation 1** *Every subset teaching sequence of order $d$ can be transformed into a* normalized *sequence $(T_k)_{k \in \mathbb{N}}$ of the same order, where a normalized subset teaching sequence has the property that, for every $k$ and every $C \in \mathcal{C}$, we have (i) $T_{k+1}$ differs from $T_k$ on exactly one concept, (ii) $|T_{k+1}(C)| \in \{|T_k(C)| - 1, |T_k(C)|\}$, (iii) $|T_k(C)| \geq d$, which implies that $|T_{k^*}(C)| = d$.*

**Proof.** Properties (i) and (ii) are easy to achieve by breaking a step from $T_k$ to $T_{k+1}$ into several smaller intermediate steps. Assume that (ii) holds. Then property (iii) can be achieved by omitting all steps that make $|T_k(C)|$ smaller than $d$. It is easy to see that the resulting sequence is again an admissible subset teaching sequence. $\qquad \square$

**Proposition 13** $\mathrm{STD}_{\min}(\mathcal{C}) \leq \mathrm{STD}(\mathcal{C})$, *and for all $n \in \mathbb{N}$ there is some succinct $\mathcal{C}_n$ such that $\mathrm{STD}_{\min}(\mathcal{C}_n) = 2$ and $\mathrm{STD}(\mathcal{C}) = n$.*

**Proof.** To see that $\mathrm{STD}_{\min}$ is bounded from above by STD, let $k^*$ be as defined in Definition 4. For each $k \leq k^*$, let $T_k(C)$ be *any one* set in $\mathrm{STS}^k(C)$ such that $T_{k^*}(C) \subseteq T_{k^*-1}(C) \subseteq \ldots \subseteq T_1(C)$. Such sets $T_k(C)$ exist by the definition of STD. Finally, set $T_0(C) = \{(x, C(x)) \mid x \in X\}$. Then $\mathcal{T} = (T_k)_{k \in \mathbb{N}}$ is a subset teaching sequence of order $\mathrm{STD}(\mathcal{C})$ for $\mathcal{C}$. So, $\mathrm{STD}_{\min}(\mathcal{C}) \leq \mathrm{STD}(\mathcal{C})$.

An example of a succinct concept class $\mathcal{C}_n$ as claimed is the class over a domain of size $n + 1$, consisting of all concepts of size either 1 or 2. It was shown by Zilles et al. [2011], that $\mathrm{STD}(\mathcal{C}) = n$. By contrast, one can easily obtain $\mathrm{STD}_{\min}(\mathcal{C}_n) = 2$, as illustrated in Table 4: for any concept $C$ of size 2, the set $T_1(C)$ contains only the two positively labeled instances for $C$, while $T_1(C) = T_0(C) = \{(x, C(x)) \mid x \in X\}$ if $C$ is a singleton. In the next iteration, set $T_2(\{x_n\}) = \{(x_n, 1), (x_1, 0)\}$ and $T_2(\{x_i\}) = \{(x_i, 1), (x_{i+1}, 0)\}$ for each singleton concept $\{x_i\}$ with $i \neq n$. Clearly, for all $i$, $T_2(\{x_i\}) \subseteq T_1(\{x_i\})$ and $T_2(\{x_i\}) \nsubseteq T_1(C)$ for any $C \neq \{x_i\}$. Thus, we obtain a subset teaching sequence of order 2 for $\mathcal{C}$, i.e., $\mathrm{STD}_{\min}(\mathcal{C}) = 2$. $\qquad \square$

| concept in $\mathcal{C}_4$ | $x_1$ | $x_2$ | $x_3$ | $x_4$ | $x_5$ | $T_1$ |
|---|---|---|---|---|---|---|
| $C_1$ | **1** | **0** | 0 | 0 | 0 | $\{(x_1, 1), (x_2, 0), (x_3, 0), (x_4, 0), (x_5, 0)\}$ |
| $C_2$ | 0 | **1** | **0** | 0 | 0 | $\{(x_1, 0), (x_2, 1), (x_3, 0), (x_4, 0), (x_5, 0)\}$ |
| $C_3$ | 0 | 0 | **1** | **0** | 0 | $\{(x_1, 0), (x_2, 0), (x_3, 1), (x_4, 0), (x_5, 0)\}$ |
| $C_4$ | 0 | 0 | 0 | **1** | **0** | $\{(x_1, 0), (x_2, 0), (x_3, 0), (x_4, 1), (x_5, 0)\}$ |
| $C_5$ | **0** | 0 | 0 | 0 | **1** | $\{(x_1, 0), (x_2, 0), (x_3, 0), (x_4, 0), (x_5, 1)\}$ |
| $C_6$ | **1** | **1** | 0 | 0 | 0 | $\{(x_1, 1), (x_2, 1)\}$ |
| $C_7$ | **1** | 0 | **1** | 0 | 0 | $\{(x_1, 1), (x_3, 1)\}$ |
| $C_8$ | **1** | 0 | 0 | **1** | 0 | $\{(x_1, 1), (x_4, 1)\}$ |
| $C_9$ | **1** | 0 | 0 | 0 | **1** | $\{(x_1, 1), (x_5, 1)\}$ |
| $C_{10}$ | 0 | **1** | **1** | 0 | 0 | $\{(x_2, 1), (x_3, 1)\}$ |
| $C_{11}$ | 0 | **1** | 0 | **1** | 0 | $\{(x_2, 1), (x_4, 1)\}$ |
| $C_{12}$ | 0 | **1** | 0 | 0 | **1** | $\{(x_2, 1), (x_5, 1)\}$ |
| $C_{13}$ | 0 | 0 | **1** | **1** | 0 | $\{(x_3, 1), (x_4, 1)\}$ |
| $C_{14}$ | 0 | 0 | **1** | 0 | **1** | $\{(x_3, 1), (x_5, 1)\}$ |
| $C_{15}$ | 0 | 0 | 0 | **1** | **1** | $\{(x_4, 1), (x_5, 1)\}$ |

Table 4: The concept class $\mathcal{C}_n$ [Zilles et al., 2011], from the proof of Proposition 13 for the case $n = 4$. The final subset teaching sets (corresponding to $T_2$) that witness $\mathrm{STD}_{\min}(\mathcal{C}_n) = 2$ are highlighted in blue. The rightmost column shows the mapping $T_1$; the subsets marked in blue are not contained in any other set in that column, hence they can be used by the teacher $T_2$ in the next iteration. When calculating STD instead of $\mathrm{STD}_{\min}$, the teacher $T_1$ assigns every singleton its unique minimum teaching set, which is a set of four negative examples. These sets cannot be reduced in subsequent iterations, since their proper subsets occur in minimum teaching sets for other concepts; hence $\mathrm{STD}(\mathcal{C}_4) = 4$.

**Proposition 15** $\mathrm{STD}_{\min}$ *is class-monotonic, domain-monotonic, and satisfies the antichain property.*

**Proof.** Class-monotonicity is obvious: If $\mathcal{C}, \mathcal{C}'$ are concept classes over a fixed domain $X$, $\mathcal{C} \subseteq \mathcal{C}'$, and $\mathcal{T}' = (T_k')_{k \in \mathbb{N}}$ is a subset teaching sequence for $\mathcal{C}'$ of order $\mathrm{STD}_{\min}(\mathcal{C}')$, then define $T_k$ to be the restriction of $T_k'$ to $\mathcal{C}$. Clearly, $\mathcal{T} = (T_k)_{k \in \mathbb{N}}$ is a subset teaching sequence for $\mathcal{C}$ of order at most $\mathrm{STD}_{\min}(\mathcal{C}')$. Hence $\mathrm{STD}_{\min}(\mathcal{C}) \leq \mathrm{STD}_{\min}(\mathcal{C}')$.

166 To establish domain-monotonicity, let $\mathcal{C}$ be any concept class over a domain $X$, and let $X' \subseteq X$
167 preserve $\mathcal{C}$. Then any subset teaching sequence $\mathcal{T}'$ for $\mathcal{C}{\downarrow}_{X'}$ can be turned into a subset teaching
168 sequence $\mathcal{T}$ for $\mathcal{C}$, by setting $T_0(C) = \{(x, C(x)) \mid x \in X\}$ and $T_k(C) = T'_k(C)$ for all $C \in \mathcal{C}$ and
169 all $k \geq 1$. Note that $\mathrm{ord}_\mathcal{C}(\mathcal{T}) = \mathrm{ord}_{\mathcal{C}{\downarrow}_{X'}}(\mathcal{T}')$. Therefore $\mathrm{STD}_{\min}(\mathcal{C}{\downarrow}_{X'}) \geq \mathrm{STD}_{\min}(\mathcal{C})$.

170 By the definition of subset teaching sequence, it is obvious that $\mathrm{STD}_{\min}$ satisfies the antichain
171 property. $\qquad\square$

## D  Proof Details for Section 5

173 **Proposition 16** *For every $n \in \mathbb{N}$ there is (i) a concept class $\mathcal{C}$ with $\mathrm{STD}(\mathcal{C}) = \mathrm{STD}_{\min}(\mathcal{C}) = 1$ and*
174 $\mathrm{NCTD}(\mathcal{C}) = n$; *(ii) a concept class $\mathcal{C}$ with $\mathrm{STD}(\mathcal{C}) = \mathrm{STD}_{\min}(\mathcal{C}) = n$ and $\mathrm{NCTD}(\mathcal{C}) = \frac{n}{2}$.*

175 **Proof.** (i) Consider the class $\mathcal{C}^{pair}_u$, as defined by Zilles et al. [2011], for any number $u \geq 3$. This
176 concept class is shown in Table 5 for $u = 3$. It is defined over $2^u + u$ instances $x_1, \ldots, x_{2^u+u}$. The
177 set $\{x_{2^u+1}, \ldots, x_{2^u+u}\}$ of the last $u$ instances is shattered. Let $\alpha_1, \ldots, \alpha_{2^u}$ be the list of all possible
178 assignments of labels to the last $u$ instances. For each such assignment $\alpha_i$, the concept class contains
179 two concepts $C_{2i-1}$ and $C_{2i}$ realizing $\alpha_i$. The concept $C_{2i-1}$ does not contain any of the first $2^u$
180 instances $x_1, \ldots, x_{2^u}$. The concept $C_{2i}$ contains $x_i$, but none of the other instances in $\{x_1, \ldots, x_{2^u}\}$.
181 See Table 5 for an illustration when $u = 3$. Note that this concept class can be equivalently written in
182 block matrix form as follows:
$$\begin{bmatrix} I_{2^u} & P_u \\ 0 & P_u \end{bmatrix}$$
183 where $P_u$ represents the powerset over a set of $u$ instances and $I_{2^u}$ is the $2^u \times 2^u$ identity matrix.

184 It was proven by Zilles et al. [2011] that $\mathrm{STD}(\mathcal{C}^{pair}_u) = 1$. We claim that $\mathrm{NCTD}(\mathcal{C}^{pair}_u) = \lceil \frac{u}{2} \rceil$.
185 To see this, note that the subclass of concepts $C_{2i-1}$, $1 \leq i \leq 2^u$ is the powerset over the last
186 $u$ instances, where all these concepts agree on the first $2^u$ instances. Thus, the NCTD of this
187 subclass equals the NCTD of the powerset over $u$ instances, which is $\lceil \frac{u}{2} \rceil$ [Kirkpatrick et al., 2019].
188 Since NCTD is class-monotonic, we have $\mathrm{NCTD}(\mathcal{C}^{pair}_u) \geq \lceil \frac{u}{2} \rceil$. A teacher mapping $T$ witnessing
189 $\mathrm{NCTD}(\mathcal{C}^{pair}_u) \leq \lceil \frac{u}{2} \rceil$ can be defined by (i) setting $T(C_{2i}) = \{(x_i, 1)\}$ for $1 \leq i \leq 2^u$, and (ii)
190 teaching the concepts $C_{2i-1}$, $1 \leq i \leq 2^u$, with a non-clashing teacher for the powerset over the last
191 $u$ instances, as used by Kirkpatrick et al. [2019]. Clearly, $T$ is clash-free.

192 For $n \in \mathbb{N}$ and $u = 2n$, thus $\mathrm{STD}(\mathcal{C}^{pair}_u) = \mathrm{STD}_{\min}(\mathcal{C}^{pair}_u) = 1$ and $\mathrm{NCTD}(\mathcal{C}^{pair}_u) = n$.

193 (ii) Consider the powerset $\mathcal{P}_n$ on $n$ instances. The fact that $\mathrm{NCTD}(\mathcal{C}) = \frac{n}{2}$ was shown by Kirkpatrick
194 et al. [2019]. It is obvious that $\mathrm{STD}_{\min}(\mathcal{P}_n) = n$: Every sample set for a concept $C \in \mathcal{P}_n$ that omits
195 one instance from $X$ is also a sample set for some concept $C' \neq C$, $C' \in \mathcal{P}_n$. Thus any subset
196 teaching sequence for $\mathcal{P}_n$ satisfies $T_k = T_0$ for all $k \in \mathbb{N}$. $\qquad\square$

## E  Details for Section 6

### E.1  Proof Details for Theorem 20

199 To complete the proof of Theorem 20, we show that $\mathrm{STD}_{\min}$ is not unambiguous on Warmuth's
200 class $\mathcal{C}_W$ which was defined by Doliwa et al. [2014] after communication with M. Warmuth. $\mathcal{C}_W$
201 is a concept class of 10 concepts over 5 instances, see Table 6. We know that $\mathrm{VCD}(\mathcal{C}_W) =$
202 $\mathrm{VCD}_{\min}(\mathcal{C}_W) = 2$, while $\mathrm{RTD}(\mathcal{C}_W) = \mathrm{STD}(\mathcal{C}_W) = 3$. It turns out that $\mathrm{STD}_{\min}(\mathcal{C}_W) \leq 2$, as
203 witnessed by the subset teaching sequence that is highlighted in Table 6. However, there is a second
204 $\mathrm{STD}_{\min}$-teacher for $\mathcal{C}_W$ that has exactly the same range as the one resulting from the subset teaching
205 sequence in Table 6 – see Table 7. A comparison of Tables 6 and 7 shows that $T_2$ and $T'_2$ swap the
206 teaching sets for $C_{2i-1}$ and $C_{2i}$, for all $i \in \{1, \ldots, 5\}$.

### E.2  Redundant Instances Can Cause Extreme Forms of Ambiguity

208 The ambiguity of $\mathrm{STD}_{\min}$ can take extreme forms for artificially created concept classes that have
209 many redundant instances. An instance $x \in X$ is redundant for $\mathcal{C}$ if $X \setminus \{x\}$ preserves $\mathcal{C}$.

| concept in $\mathcal{C}_3^{pair}$ | $x_1$ | $x_2$ | $x_3$ | $x_4$ | $x_5$ | $x_6$ | $x_7$ | $x_8$ | $x_9$ | $x_{10}$ | $x_{11}$ |
|---|---|---|---|---|---|---|---|---|---|---|---|
| $C_1$ | 0 | 0 | 0 | 0 | 0 | 0 | 0 | 0 | **0** | **0** | 0 |
| $C_2$ | **1** | 0 | 0 | 0 | 0 | 0 | 0 | 0 | 0 | 0 | 0 |
| $C_3$ | 0 | 0 | 0 | 0 | 0 | 0 | 0 | 0 | **0** | 0 | 1 |
| $C_4$ | 0 | **1** | 0 | 0 | 0 | 0 | 0 | 0 | 0 | 0 | 1 |
| $C_5$ | 0 | 0 | 0 | 0 | 0 | 0 | 0 | 0 | **0** | 1 | **0** |
| $C_6$ | 0 | 0 | **1** | 0 | 0 | 0 | 0 | 0 | 0 | 1 | 0 |
| $C_7$ | 0 | 0 | 0 | 0 | 0 | 0 | 0 | 0 | **0** | 1 | 1 |
| $C_8$ | 0 | 0 | 0 | **1** | 0 | 0 | 0 | 0 | 0 | 1 | 1 |
| $C_9$ | 0 | 0 | 0 | 0 | 0 | 0 | 0 | 0 | 1 | **0** | 0 |
| $C_{10}$ | 0 | 0 | 0 | 0 | **1** | 0 | 0 | 0 | 1 | 0 | 0 |
| $C_{11}$ | 0 | 0 | 0 | 0 | 0 | 0 | 0 | 0 | 1 | 0 | 1 |
| $C_{12}$ | 0 | 0 | 0 | 0 | 0 | **1** | 0 | 0 | 1 | 0 | 1 |
| $C_{13}$ | 0 | 0 | 0 | 0 | 0 | 0 | 0 | 0 | 1 | 1 | **0** |
| $C_{14}$ | 0 | 0 | 0 | 0 | 0 | 0 | **1** | 0 | 1 | 1 | 0 |
| $C_{15}$ | 0 | 0 | 0 | 0 | 0 | 0 | 0 | **0** | 1 | 1 | 1 |
| $C_{16}$ | 0 | 0 | 0 | 0 | 0 | 0 | 0 | **1** | 1 | 1 | 1 |

Table 5: The concept class $\mathcal{C}_u^{pair}$ [Zilles et al., 2011], for the case $u = 3$. The subset teaching sets witnessing $\mathrm{STD}(\mathcal{C}_3^{pair}) = 1$ are highlighted in blue. Non-clashing sets that witness $\mathrm{NCTD}(\mathcal{C}_3^{pair}) \leq 2$ are in bold font. The proof of Proposition 16 shows that $\mathrm{NCTD}(\mathcal{C}_3^{pair}) = 2$.

| concept | $T_0$ | | | | | $T_1$ | | | | | $T_2$ | | | | |
|---|---|---|---|---|---|---|---|---|---|---|---|---|---|---|---|
| | $x_1$ | $x_2$ | $x_3$ | $x_4$ | $x_5$ | $x_1$ | $x_2$ | $x_3$ | $x_4$ | $x_5$ | $x_1$ | $x_2$ | $x_3$ | $x_4$ | $x_5$ |
| $C_1$ | 1 | 1 | 0 | 0 | 0 | * | * | 0 | 0 | 0 | * | * | 0 | * | 0 |
| $C_2$ | 1 | 1 | 0 | 1 | 0 | 1 | 1 | * | 1 | * | 1 | 1 | * | * | * |
| $C_3$ | 0 | 1 | 1 | 0 | 0 | 0 | * | * | 0 | 0 | 0 | * | * | 0 | * |
| $C_4$ | 0 | 1 | 1 | 0 | 1 | * | 1 | 1 | * | 1 | * | 1 | 1 | * | * |
| $C_5$ | 0 | 0 | 1 | 1 | 0 | 0 | 0 | * | * | 0 | * | 0 | * | * | 0 |
| $C_6$ | 1 | 0 | 1 | 1 | 0 | 1 | * | 1 | 1 | * | * | * | 1 | 1 | * |
| $C_7$ | 0 | 0 | 0 | 1 | 1 | 0 | 0 | 0 | * | * | 0 | * | 0 | * | * |
| $C_8$ | 0 | 1 | 0 | 1 | 1 | * | 1 | * | 1 | 1 | * | * | * | 1 | 1 |
| $C_9$ | 1 | 0 | 0 | 0 | 1 | * | 0 | 0 | 0 | * | * | 0 | * | 0 | * |
| $C_{10}$ | 1 | 0 | 1 | 0 | 1 | 1 | * | 1 | * | 1 | 1 | * | * | * | 1 |

Table 6: The concept class $\mathcal{C}_W$. A subset teaching sequence can be chosen by defining $T_1(C_{2i})$ to consist of the only three positive examples for $C_{2i}$, and $T_1(C_{2i-1})$ to consist of the only three negative examples for $C_{2i-1}$, where $1 \leq i \leq 5$. In $T_2$, these sets can easily be reduced to sets of size 2. Asterisks denote instances not occurring in the chosen teaching sets.

**Example 1** *For arbitrary $n \in \mathbb{N}$, consider a concept class for which $\mathrm{VCD}$ is $n$, while $\mathrm{STD}_{\min}$ equals 1, with a large number of redundant instances. Such a class can be constructed over a domain $X$ that has $n2^n$ instances and is partitioned into $2^n$ sets $X_1, \ldots, X_{2^n}$, each of size $n$. The concept class consists of $2^n$ concepts, chosen so that they shatter each set $X_i$, $1 \leq i \leq 2^n$. See Table 8 for an illustration when $n = 2$.*

*To see that $\mathrm{STD}_{\min}$ equals 1, let $C_1, \ldots, C_{2^n}$ be an enumeration of all concepts in this concept class. It suffices to pick a teaching sequence as follows. We define $T_1(C_i) = \{(x, C_i(x)) \mid x \in X_i\}$, that means, we pick the instances in the $i$th set $X_i$ to represent the $i$th concept. Now $T_2(C_i)$ can consist of any single example from $T_1(C_i)$, since $T_1(C_i) \cap T_1(C_j) = \emptyset$ for all $j \neq i$.*

*Obviously, by reordering concepts, we obtain different $\mathrm{STD}_{\min}$-teachers that have the same range; in particular, they witness a very high degree of ambiguity, as will be formalized in Observation 1.*

Example 1 can be generalized to the following observation.

**Observation 1** *Let $\mathcal{C}$ be any concept class over a domain $X$. Suppose $X$ can be partitioned into a family $(X_C)_{C \in \mathcal{C}}$ of subsets such that $X_C$ preserves $\mathcal{C}$, for every $C \in \mathcal{C}$. Then $\mathrm{STD}_{\min}(\mathcal{C}) = 1$ and there are at least $|\mathcal{C}|!$ pairwise distinct $\mathrm{STD}_{\min}$-teachers for $\mathcal{C}$ with the same range on $\mathcal{C}$. In particular, every permutation $\sigma$ of $\mathcal{C}$ yields an $\mathrm{STD}_{\min}$-teacher that maps a concept $C$ to the singleton sample set $\{(x_{\sigma(C)}, C(x_{\sigma(C)}))\}$, where $x_{\sigma(C)}$ is any instance in $X_{\sigma(C)}$.*

| concept | $T_0' = T_0$ | | | | | $T_1'$ | | | | | $T_2'$ | | | | |
|---|---|---|---|---|---|---|---|---|---|---|---|---|---|---|---|
| | $x_1$ | $x_2$ | $x_3$ | $x_4$ | $x_5$ | $x_1$ | $x_2$ | $x_3$ | $x_4$ | $x_5$ | $x_1$ | $x_2$ | $x_3$ | $x_4$ | $x_5$ |
| $C_1$ | 1 | 1 | 0 | 0 | 0 | 1 | 1 | * | 0 | * | 1 | 1 | * | * | * |
| $C_2$ | 1 | 1 | 0 | 1 | 0 | * | * | 0 | 1 | 0 | * | * | 0 | * | 0 |
| $C_3$ | 0 | 1 | 1 | 0 | 0 | * | 1 | 1 | * | 0 | * | 1 | 1 | * | * |
| $C_4$ | 0 | 1 | 1 | 0 | 1 | 0 | * | * | 0 | 1 | 0 | * | * | 0 | * |
| $C_5$ | 0 | 0 | 1 | 1 | 0 | 0 | * | 1 | 1 | * | * | * | 1 | 1 | * |
| $C_6$ | 1 | 0 | 1 | 1 | 0 | 1 | 0 | * | * | 0 | * | 0 | * | * | 0 |
| $C_7$ | 0 | 0 | 0 | 1 | 1 | * | 0 | * | 1 | 1 | * | * | * | 1 | 1 |
| $C_8$ | 0 | 1 | 0 | 1 | 1 | 0 | 1 | 0 | * | * | 0 | * | 0 | * | * |
| $C_9$ | 1 | 0 | 0 | 0 | 1 | 1 | * | 0 | * | 1 | 1 | * | * | * | 1 |
| $C_{10}$ | 1 | 0 | 1 | 0 | 1 | * | 0 | 1 | 0 | * | * | 0 | * | 0 | * |

Table 7: A second subset teaching sequence for the concept class $\mathcal{C}_W$.

| concept | $X_1$ | | $X_2$ | | $X_3$ | | $X_4$ | |
|---|---|---|---|---|---|---|---|---|
| | $x_1$ | $x_2$ | $x_3$ | $x_4$ | $x_5$ | $x_6$ | $x_7$ | $x_8$ |
| $C_1$ | 0 | 0 | 0 | 0 | 0 | 0 | 0 | 0 |
| $C_2$ | 0 | 1 | 0 | 1 | 0 | 1 | 0 | 1 |
| $C_3$ | 1 | 0 | 1 | 0 | 1 | 0 | 1 | 0 |
| $C_4$ | 1 | 1 | 1 | 1 | 1 | 1 | 1 | 1 |

Table 8: The concept class from Example 1, for the case $n = 2$. Highlighted in blue are the labels chosen for teaching individual concepts with $T_1$. Clearly, $T_2$ can be defined to assign each concept a singleton sample set.