# OpenReview forum: "On Batch Teaching with Sample Complexity Bounded by VCD"
_NeurIPS.cc/2022/Conference — NeurIPS 2022 Accept_

### Official Review · Reviewer_hi6B · 2022-07-06

**Rating:** 7
**Confidence:** 3
**Soundness:** 2 fair
**Presentation:** 3 good
**Contribution:** 2 fair

**Summary:**

In this paper the authors make progress towards the question of relating passive learning and active learning. A long-standing open question in this area is the relation between recursive teaching dimension (RTD), which in a way characterizes the number of labelled examples sufficient to learn a concept class, and VC dimension (VC) which characterizes the number of random labelled examples sufficient to PAC learn the concept class. The best known upper bound in this direction is RTD < VC^2 for every concept class and we do know of some small-sized concept classes (say on 10 bits) for which RTD >= 2 VC. The opimal relation remains open and has applications to sample compression of learning.

In a single line, the main result of the paper is proposing a new combinatorial quantum STD_min, which is smaller than RTD and for this new quantity they show that STD_min <= VC.

The main strength of this paper is, the authors have been able to understand this property called GM Collusion freeness (here GM stands for initials of authors in a prior paper) which is a property of learning. In this direction there has been one combinatorial parameter  NCDIM which satisfies GM collusion freeness and its open if NCDIM <= VC. It was so far believed that GM collusion freeness is a necessary property in order to for a parameter to be upper bounded by VC. In this paper the parameter constructed by the authors STD_min is  not GM-collusion-free and yet is upper bounded by VC dimension. I find this a cute structural property that the authors have observed.

As for weakeness, what's unclear to me is, couple of things
a) How artificial is STD_min? In the sense, is it such a restricted model of active learning which *allows* to be upper bounded by VC dimension or is there something natural about this combinatorial parameter?
b) One of the solid motivations of understanding RTD versus VC is sample compression schemes for learning. Its not at all clear to me if this new relation of STD_min < VC can help understand sample compression or not.

With this in mind, I think this paper would be a boderline/weak accept for NeurIPS: it makes a nice contribution in a hard open question but the main contribution might be overall contrived and not help resolve the open question completely.

**Questions:**

Discussed above.

**Ethics Review Area:**

["Responsible Research Practice (e.g., IRB, documentation, research ethics)"]

**Limitations:**

Societal impact of this work is not applicable. It is a theory paper.

**Strengths And Weaknesses:**

Discussed above.

---

> ### Author Response · Authors · 2022-07-30
> **Response to Reviewer hi6B**
>
> Thank you for carefully reviewing our paper! We greatly appreciate your feedback. Please see below our responses to your comments.
>
> -----
> **1. How artificial is STD_min? In the sense, is it such a restricted model of active learning which allows to be upper bounded by VC dimension or is there something natural about this combinatorial parameter?**
> - In an abstract setting, devoid of any specific application, it is not always easy to talk about the naturalness of learning/teaching models. It may therefore be helpful to consider various properties of teaching that one might find desirable and then to design models that possess (some of) these properties. We have displayed some such properties, which are fulfilled by STD_min, in Table 2 in the conclusions. Another property of STD_min, that is not fulfilled by TD, RTD or NCTD, is that it allows the learner to discard a hypothesis because the given data is not *important* for that hypothesis (even if it is consistent). We argue that a learner who knows that data comes from a helpful teacher, has reasons to reject hypotheses for which the data is not important/helpful. Of course, there may be other natural properties that are *not* fulfilled by STD_min. An interesting research question is what natural properties can (or cannot) be fulfilled by teaching models whose parameter is upper-bounded by VCD.
>
> -----
> **2. One of the solid motivations of understanding RTD versus VC is sample compression schemes for learning. Its not at all clear to me if this new relation of STD_min < VC can help understand sample compression or not.**
> - Coincidentally, Zachary Chase recently published on ArXiv (“Optimally Compressing VC Classes”, arXiv:2201.04131, 2022) a proof attempt for the sample compression conjecture that used a technique very similar to the one used by (Mansouri et al. 2019) and which we deployed for our proof that STD_min <= VCD. Chase later withdrew his technical report, due to a flaw in a lemma, but apparently he too sees some value in the proof technique that we used, specifically in the context of sample compression. If our proof is published, it increases the chances that someone can use it in this context. Note also that results on teaching dimension parameters have been used to obtain new results on sample compression in the past. One example is the paper “Order Compression Schemes” by Darnstadt et al. (Theoretical Computer Science 620:73-90, 2016), which uses results on the Recursive Teaching Dimension in order to obtain new forms of sample compression schemes. It lies in the very nature of research that only some of the published techniques will end up being used, but unless one is certain that our proof technique is useless beyond our paper, there may indeed be value in publishing it.
>
> -----
>
> We hope that our responses have addressed your concerns. If you have any other comments or feedback, please let us know. Thank you again for the review!

---

### Official Review · Reviewer_Rn8w · 2022-07-10

**Rating:** 8
**Confidence:** 3
**Soundness:** 3 good
**Presentation:** 4 excellent
**Contribution:** 4 excellent

**Summary:**

The paper discusses the model of batch teaching, and seeks a reasonable batch teaching complexity measure that is upper bounded by a linear function of the VC dimension, which is a parameter that controls the sample complexity of learning in the PAC model.

First of all, there is a nice review of known teaching dimensions suggested in previous papers - TD, RTD, NCTD, STD, and their properties.
Then, the authors suggest 3 properties that a "reasonable" teaching model should satisfy - class-monotonicity, domain-monotonicity, and antichain property.
Moreover, the suggested parameter should violate criteria called "GM-collusion-free", in order to be of size linear in the VC dimension.

Following the above, the authors suggest a parameter called STD_min (and explain why the other parameters "fail"). This parameter satisfies the 3 criteria and is of the size of the VC dimension. On the negative side, this parameter loses a desirable property of unambiguity.

**Questions:**

-

**Ethics Review Area:**

["I don’t know"]

**Limitations:**

-

**Strengths And Weaknesses:**

The paper (which is full of technical details) is clearly presented.

It is well-motivated why one would search for a new teaching parameter, by carefully reviewing the known parameters and their properties.

I think that there are potential implications of the ideas in this paper on learning and teaching.

Overall I find this paper very elegant and I recommend it for acceptance.

---

> ### Author Response · Authors · 2022-07-30
> **Response to Reviewer Rn8w**
>
> Thank you for carefully reviewing our paper! We greatly appreciate your feedback. If you have any other comments or feedback, please let us know.

---

### Official Review · Reviewer_nVaX · 2022-07-11

**Rating:** 5
**Confidence:** 1
**Soundness:** 3 good
**Presentation:** 3 good
**Contribution:** 2 fair

**Summary:**

The authors present a linear bound in terms of Vapnic-Chevronenkis
dimension, or VCD, for a model of batch "teacher-student"
learning. Note that in machine teaching is not a machine learning
problem: the question is for a concept, what is the smallest sample
size (from what I gathered, from a tutorial reference). This
information can still be useful in understanding difficulty of
teaching or learning of various concept classes. Previous linear
bounds were for sequential and not batch models of interaction. One
challenge is how to formulate the constraints so that the teacher and
student cannot collude (from my understanding, allowing a teacher to
potentially cheat and somehow encode information about the target
concept to the student). In this model of batch-teacher learning, the
authors' result is to have shown the first linear bound in terms of
VCD (previous results being Quadratic for the batch case, and only linear for the sequential).


**Questions:**

-- the first mention of 'coding tricks' in the introduction was
 unclear to me (a researcher who hadn't seen this framework). The next
 sentence and rereading the firt few pages give me an idea. You can
 preface by the 'coding tricks' sentence by one that .

-- My first impression: Shouldn't teaching lower the bound from VCD?
 (instead of being quadratic or even linear) Since the teacher is
 trying to be helpful! After reaching the paper, and a bit of the
 citations, I gather here the requirement in the problem formulation is higher, and the problem
 is not even an ml problem! (but some inverse of it!)

-- Preliminaries section: please reemphasize here that the definitions
 are for batch learning. (I assume that's the case, since sets of
 examples are provided from teacher to student)

-- For a more general audience, would be good to provide some concrete
 examples of what a concept class can be, etc. early in paper.



**Limitations:**

Not applicable.

**Strengths And Weaknesses:**

Strengths:

--  Clarity of defining the problems, and the improvements in the bound.

Weakness:

-- The paper is highly abstract and the work is completely
 theoretical.  A theoretical machine learning venue is more
 appropriate.

-- Their notion of teachability, for a concept class, seems contrived,
 and, again, appears be of purely theoretical interest, currently.

---

> ### Author Response · Authors · 2022-07-30
> **Response to Reviewer nVaX**
>
> Thank you for carefully reviewing our paper! We greatly appreciate your feedback. Please see below our responses to your comments.
>
> -----
> **1. The paper is highly abstract and the work is completely theoretical. A theoretical machine learning venue is more appropriate.**
> - The broad machine learning scope of NeurIPS does include purely theoretical work as well. For example, the paper (Mansouri et al. 2019) that we cite in our work was published in NeurIPS. Likewise, Wang et al. published “A mathematical theory of cooperative communication” in NeurIPS 2020. In 2016, Chen et al. published “On the Recursive Teaching Dimension of VC Classes” in NeurIPS. These three papers are quite similar in style to ours, and they all deal with machine teaching. In addition, there are many NeurIPS papers on computational learning theory that do not deal with machine teaching, yet are similarly abstract and theoretical as the ones mentioned here.
>
>
> -----
> **2. the first mention of 'coding tricks' in the introduction was unclear to me (a researcher who hadn't seen this framework).**
> - Thank you for pointing this out. This will be easy to address when revising the paper.
>
> -----
> **3. Preliminaries section: please reemphasize here that the definitions are for batch learning.**
> - You are correct; we will emphasize this more strongly in our revision.
>
> -----
> **4. For a more general audience, would be good to provide some concrete examples of what a concept class can be, etc. early in paper.**
> - That is a very useful suggestion; we will address this in the revised paper.
>
> -----
>
> We hope that our responses have addressed your concerns. If you have any other comments or feedback, please let us know. Thank you again for the review!

---

> > ### Comment · Reviewer_nVaX · 2022-08-08
> > **Read the author responses**
> >
> >
> > I have read all the reviews and the author responses.   My questions and comments have been adequately addressed.
> > I remain borderline on the contributions of the paper, and I am happy if it is accepted to NeurIPS.
> >
> > Thank you.

---

> > > ### Author Response · Authors · 2022-08-09
> > > **Response to comment**
> > >
> > > Thank you for your response to our rebuttal. We are glad to read that you are not opposed to acceptance of the paper, and hope that our responses are helpful in improving your rating. If you have any other comments or feedback, please let us know. Thank you again for your careful review.

---

### Official Review · Reviewer_47WT · 2022-07-18

**Rating:** 7
**Confidence:** 2
**Soundness:** 4 excellent
**Presentation:** 3 good
**Contribution:** 4 excellent

**Summary:**

The paper investigates the problem of machine teaching - the ability to cast the interaction by a teacher (who is using labelled instances to teach a concept) and a learner in a way that obtains small complexity in terms of the required labelled examples. The authors suggest and analyzes a new batch teaching complexity measure they term STD_min, and show that while having several desirable properties (instead of the standard in the literature requirement of collusion-freeness), it may be upper bounded by the VC-dimension of the concept class being taught. In doing so, the paper provides the first measure of teaching complexity that is bounded (or is even linear in) the VC-dimension of the concept class.

**Questions:**

-

**Limitations:**

Limitations are adequately discussed in the conclusions section.

**Strengths And Weaknesses:**

Originality:

Strengths:
1. The problem studied by the paper is interesting and fundamental.
2. The work suggests a new measure of the teaching complexity, while showing it has desirable properties (which replace the standard in the literature collusion-freeness assumption, for which the authors also provide practical motivation), showing an upper bound of it by the VC-dimension.

Quality:
The submission appears to be technically sound. Relevant related work is included in context.

Clarity:
The submission is very well written.

Significance:
Overall I feel this is a strong paper. I do not actively conduct research in this sub-area, so very accurate assessment of the impact is a bit difficult for me. However, by reading and understanding the related work as it appears in the submission, and the suggested contributions, the results seem novel and significant.

---

> ### Author Response · Authors · 2022-07-30
> **Response to Reviewer 47WT**
>
> Thank you for carefully reviewing our paper! We greatly appreciate your feedback. If you have any other comments or feedback, please let us know.

---

> > ### Comment · Reviewer_47WT · 2022-08-07
> > **Thank you for the response**
> >
> > After reading the response from the authors, my assessment of this work remains unchanged.

---

### Author Response · Authors · 2022-07-30
**Response to all reviewers**

We thank all the reviewers for their careful reviews. Below, we provide responses to each reviewer separately.

---

### Meta-Review · Area_Chair_1Z9R · 2022-08-25

**Recommendation:** Accept
**Confidence:** Certain

**Metareview:**

The reviewers agree that this is a solid contribution. Please do revise the paper according to the reviewers comments and the discussion.

**Award:**

No

---

### Decision · Program_Chairs · 2022-09-14

Accept